# Glacial state of the global carbon cycle: time-slice simulations for the last glacial maximum with an Earth-system model

Takasumi Kurahashi-Nakamura[1], André Paul[1], Ute Merkel[1], and Michael Schulz[1]

[1]MARUM - Center for Marine Environmental Sciences and Faculty of Geosciences, University of Bremen, Bremen, Germany

**Correspondence:** Takasumi Kurahashi-Nakamura (tkurahashi@marum.de)

**Abstract.** Three time-slice carbon-cycle simulations for the last glacial maximum (LGM) constrained by the $CO_2$ concentration in the atmosphere and the increase in the mean concentration of dissolved inorganic carbon in the deep ocean were carried out with a fully-coupled comprehensive climate model (the Community Earth System Model version 1.2). The three modeled LGM ocean states yielded different physical features in response to artificial freshwater forcing, and, depending on the physical states, suitable amounts of carbon and alkalinity were added to the ocean to satisfy constraints from paleo-data. In all the simulations, the amount of carbon added was in line with the inferred transfers of carbon among various reservoirs during the evolution from the LGM to the pre-industrial (PI) period, suggesting that the simulated glacial ocean states are compatible with the PI one in terms of the carbon budget. The increase in total alkalinity required to simulate ocean states that were deemed appropriate for the LGM was in broad quantitative accord with the scenario of post-glacial shallow water deposition of calcium carbonate, although a more precise assessment would demand further studies of various processes such as the land chemical weathering and deep-sea burial of calcium carbonates, which have affected the alkalinity budget throughout the history since the LGM. On the other hand, comparisons between the simulated distributions of paleoceanographic tracers and corresponding reconstructions clearly highlighted the different water-mass geometries and favored a shallower Atlantic meridional overturning circulation (AMOC) for the LGM as compared to PI.

## 1 Introduction

Global climate change during the last 800 kyr was characterized by the periodic variations of ice-sheet volume and atmospheric $CO_2$ concentration ($pCO_2$) with a $\sim$100-kyr period (e.g. Petit et al., 1999; Augustin et al., 2004), i.e., the glacial-interglacial cycles. The synchronous behaviors of those two quantities imply a strong link between them, and it is suggested that the variations in $pCO_2$ at the glacial-interglacial timescale were regulated by the evolution of ice sheets that represented a nonlinear response of the Earth system to the orbital forcing (e.g. Ganopolski and Brovkin, 2017). This mechanism for the $pCO_2$ variations is also supported by the fact that the periodic changes in ice-sheet volume can be numerically simulated even under temporally constant $pCO_2$ if it is sufficiently low (e.g. Ganopolski and Calov, 2011; Abe-Ouchi et al., 2013). Therefore, to comprehend the mechanism of $pCO_2$ variations in the glacial-interglacial cycles, it is fundamental to understand the response of the global carbon cycle to a given change in the states of other climatic components that work as external factors from a carbon-cycle viewpoint.

Ideally, to grasp the $pCO_2$ variations over an entire glacial cycle, a time-dependent or transient framework with a fully-coupled comprehensive climate model would be beneficial to explicitly deal with a gradually changing background state regulated by slow processes that work at a $10^4$-year timescale such as the changes of the orbital configuration and the growth of ice sheets. From the perspective of the marine carbon cycle, which is considered to have played a major role in the global carbon cycle (e.g., Sigman and Boyle, 2000), the changes in the inventory of total alkalinity as well as carbon in the entire ocean would become relevant for the dynamics at such a $10^4$-year timescale. The changes in the alkalinity inventory is considered to have occurred as the result of a slowly-accumulated imbalance in alkalinity budgets through two different processes as follows.

First, the global sea-level changes accompanied by the growth or retreat of ice sheets affect the area of continental shelves covered by seawater, which is of relevance for the shallow-water ecosystem. For example, the sea level has risen since the LGM and seawater has flooded the continental shelves, so that the modern-day area of shallow water environments for carbonate deposition including coral reef buildup is three times as large as that for the LGM (Cartapanis et al., 2018). As a result, a substantial amount of alkalinity is considered to have been removed from the LGM ocean by the post-glacial shallow water deposition of $CaCO_3$ and it is estimated that this process would account for the removal of total alkalinity by as much as 0.44–3.6 $\times 10^{17}$ eq (Milliman, 1993; Opdyke, 2000; Ridgwell et al., 2003; Vecsei and Berger, 2004; Husson et al., 2018; Köhler and Munhoven, 2020). Because the removal of alkalinity will cause the increase of the partial pressure of $CO_2$ in the surface ocean, a hypothesis that explains a substantial part of the glacial-interglacial $pCO_2$ variation by the shallow water deposition of $CaCO_3$ was proposed (the so-called "coral reef hypothesis"; e.g. Berger, 1982; Opdyke and Walker, 1992). Although this hypothesis inevitably requires that the $pCO_2$ rise followed the expansion of the area of shallow water deposition, the actual deglaciation had the opposite sequence; that is to say, a substantial part ($\sim$70 ppm) of the recorded $pCO_2$ increase happened during the first ten thousand years following the LGM (Bereiter et al., 2015; Köhler et al., 2017), while the expansion of the shallow water environments is estimated to have mostly occurred afterward (Cartapanis et al., 2018). Alternatively, it has been demonstrated with a simplified carbon cycle model that the post-glacial buildup of shallow-water carbonates is consistent with the observed $\sim$20-ppm rise of $pCO_2$ during the late Holocene (Ridgwell et al., 2003).

Second, the total inventory of alkalinity is subject to change also through the deposition or dissolution of $CaCO_3$ in the deep sediments of the open ocean, because the amount of alkalinity in the ocean is affected by a long-term budget of $CaCO_3$ through its input by chemical weathering on land and the removal by sedimentary burial at the ocean floor. In particular, if there is an imbalance between the input and removal, the alkalinity inventory will respond so that the flux balance becomes restored at a timescale of $10^4$ years (e.g. Boudreau et al., 2018). For example, a positive imbalance (i.e. more input than removal) would be removed by an increased $CaCO_3$ deposition, because seawater with a higher alkalinity concentration is more favorable to the deposition of $CaCO_3$. This intrinsic negative feedback process by the ocean is known as "carbonate compensation", which has been considered to be a major mechanism affecting a long-term climate evolution (e.g. Broecker and Peng, 1987; Archer et al., 2000; Brovkin et al., 2007; Chikamoto et al., 2008; Lord et al., 2016; Boudreau et al., 2018; Kobayashi et al., 2021).

Although transient simulations including those "slow" processes are advantageous to cope with the evolution of the global carbon cycle spanning the glacial-interglacial timescale, the computational cost for such long-term simulations with a comprehensive climate model is still prohibitive. Instead, a more practical way is to conduct time-slice simulations, for example, for

the LGM, that are subject to reconstructed boundary conditions such as the greenhouse-gas concentrations and orbital configuration. Such time-slice simulations may also serve as an initial state for shorter transient runs that cover a part of the full glacial cycle (e.g. the last deglaciation) instead of a "true" LGM state that results from the evolution of the global carbon cycle since (at least) the last interglacial period.

The main purpose of this study is to carry out fully-coupled LGM time-slice simulations including the global carbon cycle. In our model runs, each of which had 2500 model-years, we fixed the ice-sheet configuration, the orbital configuration, and the inventory of total alkalinity in the ocean as external factors or a background field assuming that the effects of their variations were minor at a millennial timescale. Instead, processes that govern the marine carbon cycle at shorter timescales such as air–sea gas exchange, the biological pump and the change of tracer distributions by ocean circulation (e.g. Hain et al., 2014)

were explicitly modelled and simulated. Therefore, the simulations in this study inevitably do not capture the influence of long-term trends in the boundary conditions. For example, the properties of very old water do not explicitly reflect the past state of boundary conditions but only approximate the true properties.

To constrain the carbon-cycle state during the LGM, we used two important restrictions in terms of the size of carbon reservoirs: atmospheric $pCO_2$ and an estimated rise of the mean concentration of dissolved inorganic carbon (DIC) in the deep

ocean. The LGM $pCO_2$ level is known to be approximately 190 ppm based on ice-core records (Bereiter et al., 2015; Köhler et al., 2017). The increase in DIC concentrations in the deep ocean has been estimated based on an empirical linear relationship between the $^{14}C$ ventilation age and modern concentrations of DIC below a depth of 2000 m, suggesting that the average DIC concentrations in the abyss were higher during the LGM by 85–115 $\mu$mol/kg (corresponding to 730-980 GtC; Sarnthein et al., 2013) or 82 $\mu$mol/kg (687 GtC; Skinner et al., 2015) as compared to the modern-day. We adjusted the mean concentrations of

DIC and total alkalinity in the entire ocean to satisfy the two constraints in addition to adopting the PMIP4 protocol regarding LGM boundary conditions (Kageyama et al., 2017).

The increase in deep-water DIC storage during the LGM should be the outcome of two combined factors: the change in the vertical distribution of stored carbon and the change in the mean concentration. The increased vertical contrast contributes to the decrease in $pCO_2$ as the increase of alkalinity does, while the increase of the mean DIC concentration has the opposite

effect. The adjustment of DIC concentrations for the entire ocean in this study will indicate whether the net effect of these two factors is in line with the estimated increase in the deep DIC concentrations.

We thereby evaluate the LGM marine carbon cycle from the perspective of the specific change in the pH of seawater that is compatible with the low-$pCO_2$ atmosphere. The carbonate ion concentration $[CO_3^{2-}]$ in the deep ocean is tightly linked to processes that affect the redistribution of carbon in the ocean and thus is a valuable tracer for investigating the global carbon

cycle (e.g., Yu et al., 2014). The simulated $[CO_3^{2-}]$ and the mass accumulation rate (MAR) of $CaCO_3$ at the ocean floor are both compared to reconstructions. We also discuss the compatibility between the modeled LGM and modern states in terms of the carbon and alkalinity budget. That is to say, we examine the likelihood that we can reconcile the total inventory of DIC and alkalinity of the modeled LGM states with the modern counterparts taking into account the changes in the sizes of several reservoirs during the transition from the LGM to PI.

## 2 Methods

### 2.1 Models

We used the Community Earth System Model (CESM; Hurrell et al., 2013) version 1.2, which is an atmosphere-ocean-land coupled comprehensive Earth-system model. The ocean component, the Parallel Ocean Program version 2 (POP2), was config-
ured to include the Biogeochemical Elemental Cycling model (BEC; Moore et al., 2004, 2013; Lindsay et al., 2014). The BEC included a nutrient–phytoplankton–zooplankton–detritus (NPZD) marine ecosystem model and handled sinking processes of biological particles to compute the distribution of various biogeochemical tracers such as phosphate, apparent oxygen utiliza-tion (AOU), and carbonate ion concentration. For the sinking of particulate organic matter (POM), the "ballast effect" was taken into account based on Armstrong et al. (2002). The ocean component was extended by the carbon-isotope package de-
veloped by Jahn et al. (2015) so that we were able to explicitly simulate the carbon-isotope composition of seawater. The land component (the Community Land Model; CLM4.0) included a prognostic treatment of the terrestrial carbon and nitrogen cy-cles to calculate the reservoir size of land carbon. A low-resolution configuration of CESM was used in this study to reduce the computational cost (Shields et al., 2012); that is to say, the ocean component had a nominal $3°$ irregular horizontal grid with 60 vertical levels, while the atmosphere component, Community Atmosphere Model version 4 (CAM4), had a T31 spectral
dynamical core (horizontal resolution of $3.75°$) with 26 vertical levels.

In this study, the size of each carbon reservoir including the atmospheric $p\mathrm{CO}_2$ is calculated by the carbon cycle model. However, we switched off the feedback from the carbon cycle to the climate system by prescribing a constant $p\mathrm{CO}_2$ that determined the radiative forcing. We treated the marine biogeochemical system as being "closed" to exclude the drift of the total inventory from prescribed amounts, that is, there were no fluxes of biogeochemical matter through the top and bottom
boundaries of the ocean domain except for the air-sea gas exchange. Particulate matter that reached the ocean floor was assumed to be remineralized in the bottom-most cells. To enhance the numerical stability, we replaced the original pH solver in POP2 with SolveSAPHE (Solver Suite for Alkalinity–pH Equations) developed by Munhoven (2013). We introduced this pH solver originally to avoid numerical instability especially in test runs or in a spin-up phase and continued to use it for consistency. We took into account the alkalinity that arises from the dissociation of the solvent water itself in addition to the carbonate and
borate alkalinity, which provided a sufficiently accurate solution (Munhoven, 2013).

For diagnostic purposes, we adopted the Model of Early Diagenesis in the Upper Sediment of Adjustable complexity (MEDUSA(v.2); Munhoven, 2020) to explicitly calculate the preservation or dissolution of $\mathrm{CaCO}_3$ at the ocean floor that corresponded to each of our CESM simulations. MEDUSA is a one-dimensional advection-diffusion-reaction model that de-scribes the early diagenetic processes involving carbonates and organic matter (OM) in the surface sediment of 10 cm thickness
as a function of time. The configuration of MEDUSA was the same as in our previous application (Kurahashi-Nakamura et al., 2020) except for diffusive boundary layers (DBL) to better represent the solute fluxes across the water-sediment boundaries (Munhoven, 2020) in this study.

## 2.2 Experiments and analyses

We performed four main time-slice experiments: one control experiment that corresponds to the PI reference period and three experiments with the LGM as a target (Table 1). For the PI run (expPI) we followed the DECK (Diagnostic, Evaluation and Characterization of Klima) PI control protocol (Eyring et al., 2016) for the physical forcing to CESM. The control experiment was initialized from the final state of the spin-up run of Kurahashi-Nakamura et al. (2020) and run for 2500 model years. For the model parameters, we followed the default settings of CESM1.2 (https://www.cesm.ucar.edu/models/cesm1.2/cesm/doc/modelnl/modelnl.html, last access: 01 August 2022).

For the LGM runs, we followed the experimental design established by the Paleoclimate Modelling Intercomparison Project Phase 4 (PMIP4; Kageyama et al., 2017). We set the radiative and orbital forcing parameters as specified in this protocol, and adjusted the ice sheet configuration based on the ICE-6G-C reconstruction (Argus et al., 2014; Peltier et al., 2015). The dust and other aerosol forcings of the atmosphere and marine biogeochemistry are based on Albani et al. (2014). The mean salinity and nutrient concentrations of the ocean were modified according to the change in ocean volume (Kageyama et al., 2017). Namely, we increased the salinity by 1 psu and all other tracer concentrations by 3%.

For the baseline LGM experiment (expLGM), we carried out a 2500-year fully-coupled carbon-cycle run which was branched from a separate physical spin-up without the marine biogeochemistry. We also conducted two sensitivity experiments with the same length for the LGM (expLGMws and expLGMss) with additional freshwater forcing to examine the dependency of the biogeochemical state on the physical ocean state. For the first sensitivity experiment (expLGMws), 0.1 Sv additional freshwater in total was uniformly added to the high-latitude North Atlantic Ocean throughout the simulation. The southern boundary of the freshwater-forcing region was defined at 50N$^\circ$, and the others by pre-defined CESM1.2 region masks (Fig. 1). For the second sensitivity experiment (expLGMss), we extracted 0.1 Sv from the same high-latitude region in the North Atlantic Ocean and 0.25 Sv in total from the Weddell Sea and Ross Sea regions. The primary motivation for the additional freshwater forcing was to prepare different physical ocean states as in previous studies (e.g., Gu et al., 2020; Muglia and Schmittner, 2021), and the freshwater amount was determined empirically to realize an AMOC with significantly different characteristics compared to the baseline state. Implicitly the additional freshwater forcing takes into account the uncertainty in the glacial freshwater budget in high latitudes due to ice-sheet calving and iceberg transport and melting processes (Merino et al., 2016) that were not explicitly modelled in our framework. The total sea-water volume and hence the global-mean concentrations of tracers were conserved by adding uniform compensating fluxes of the opposite sign over the rest of the global ocean.

The mean concentration of DIC in the LGM simulations was set based on an independent proxy-based estimate of the increase in the concentration in the deep ocean by 100 $\mu$mol/kg as compared to the modern value by Sarnthein et al. (2013) (Table 1). In fact, the increment of the DIC concentration for our three individual LGM simulations was determined through corresponding preparatory simulations (Fig. 2). To initialize the preparatory runs, we increased the global mean DIC concentration homogeneously by 100 mmol/m$^3$ compared to the PI amount as a first guess (Table 2). Depending on the respective LGM simulations, we also added 75–100 meq/m$^3$ of total alkalinity on top of the ocean-volume effect to tune the model so that a $p$CO$_2$ level between 180 ppm and 190 ppm was reached. After a 1000-yr model integration for each simulation, we

calculated the resulting mean DIC anomalies in the ocean deeper than 2000 m in accordance with Sarnthein et al. (2013). To set the initial mean concentration of DIC for the main LGM simulations, we took the difference between each of the posterior DIC anomalies of the preparatory runs and the observation-based estimate and adjusted the first guess (i.e. 100 mmol/m$^3$) by adding the difference. Thereby we obtained second-guess values that were used to initialize the respective main LGM runs (Table 1) so that we were able to achieve the DIC anomalies in the deep ocean in better accordance with the estimate by Sarnthein et al. (2013). For the main LGM runs, we newly adjusted the total alkalinity independently of the preparatory runs. 40 meq/m$^3$ to 80 meq/m$^3$ of alkalinity depending on the experiments was added in addition to the increase due to the change in the seawater volume (Table 1). The increment was chosen similarly such that the model predicted an atmospheric $p$CO$_2$ level of approximately 190 ppm based on ice-core records (Bereiter et al., 2015; Köhler et al., 2017).

The initial $\delta^{13}$C of DIC was uniformly zero, and we assumed the $\delta^{13}$C value of the atmosphere during the LGM to be -6.46‰ (Schmitt et al., 2012). Although the treatment of $^{13}$C in the current model configuration is not self-consistent because the air–sea gas exchange should have affected the atmospheric $\delta^{13}$C in reality, we took the approach for three practical reasons. 1. The prescribed atmospheric $\delta^{13}$C set to a reliable observed value was expected to result in a better model representation of $\delta^{13}$C$_{DIC}$ in the ocean that could be compared with observation-based data. 2. The deviation of the atmospheric $p$CO$_2$ from the required 190 ppm was very small, so that the simulated state would approximate the consistent LGM state in a reasonable way, including the atmospheric $\delta^{13}$C value. 3. As far as we recognize, the available carbon isotope package (Jahn et al., 2015) does not deal with atmospheric $\delta^{13}$C that evolves interactively and self-consistently with the air–sea gas exchange in the model, and therefore, a further substantial model-development will be needed to implement it, which is beyond the scope of this study.

For each experiment, we also diagnosed and analyzed upper-sediment properties with regard to CaCO$_3$ by running MEDUSA separately as a stand-alone model, taking the appropriate boundary conditions from the corresponding CESM state. As in Kurahashi-Nakamura et al. (2020), one MEDUSA column was coupled to the deepest grid cell of each POP2 water column and there was no lateral exchange of information among the MEDUSA columns.

It should be noted that the experiments in this study had a different total amount of carbon stored in the whole (atmosphere–ocean–land) system, reflecting the uncertain physical ocean states. In our experiments, instead of fixing the total amount, we tuned the size of the atmospheric reservoir to a specific amount (i.e. ~190 ppm), which practically governed the terrestrial reservoir's size, and also tuned the size of the deep-ocean reservoir as well. The size of the shallow-ocean reservoir differed among the different experiments depending on the vertical gradient of DIC concentration in the ocean, hence the ocean circulation.

## 3 Results

### 3.1 Physical ocean states

The different forcing factors and boundary conditions used for the PI and LGM time-slices yielded four very distinct physical ocean states. As expected, there were noticeable differences in the global ocean circulation and water-mass distributions. The standard LGM experiment (expLGM) had a stronger Atlantic meridional overturning circulation (AMOC) with a similar depth

structure compared to that of the pre-industrial run (expPI) (Fig. 3). Both experiments with the additional freshwater forcing resulted in noticeably shallower AMOCs, which shoaled by 500–1000 m in terms of the position of the zero-flux level that separates the upper and lower circulation cells (Fig. 3). On the other hand, the maximum strength of the upper circulation in expLGMss was stronger than in expPI, while that for expLGMws was weaker. In short, expLGMss had a stronger but shallower AMOC compared to PI, while expLGMws had a weaker and shallower AMOC. Both experiments showed stronger bottom circulations than expPI or expLGM.

The differences in the annually-averaged global-mean sea surface temperature (SST) between the respective LGM experiment and expPI were $-2.6$ K (expLGM), $-2.7$ K (expLGMss), and $-2.0$ K (expLGMws). Those values were within the range of various estimates of the mean SST anomaly for the LGM (Annan and Hargreaves, 2013; Kurahashi-Nakamura et al., 2017; Tierney et al., 2020; Paul et al., 2021) implying that the solubility of $CO_2$ in the model is consistent with the LGM climate. The vertical gradient of modeled LGM salinity was larger than in expPI in general (i.e. comparatively more saline in the very deep ocean) except for the Atlantic Ocean in expLGMws (Fig. 4e-j), which corresponded to more stratified ocean states. However, the degree of stratification differed depending on the magnitude of the additional freshwater forcing. Accordingly, the horizontal distributions of the bottom salinity varied among the LGM experiments (Fig. 4a-g). The best fit to the reconstructed salinity of bottom water at several locations (Adkins et al., 2002; Insua et al., 2014; Homola et al., 2021) was obtained in expLGMss that had the highest bottom salinity, suggesting that the more vigorous penetration of more saline southern-sourced water contributed to the result. In contrast, in expLGMws, the ocean was just slightly more stratified compared to expPI on the whole, and the model–data misfits were the largest.

## 3.2 Carbon reservoirs

The $pCO_2$ predicted by the carbon cycle module of the model was in a quasi-steady state during at least the last 1000 model years in all the experiments, and the drift of $pCO_2$ in the last 500 years was 0.6 ppm or less (depending on the simulations). In expPI that served as the control case, the predicted $pCO_2$ was 276 ppm (Table3), which agreed well with the prescribed radiative $pCO_2$ that was used to force the model climate (280 ppm). All the LGM runs reached a $pCO_2$ in the range between 180 ppm and 190 ppm as aimed for, which was approximately 90 ppm lower than in expPI and thus in good agreement with the $pCO_2$ difference obtained from ice cores (Bereiter et al., 2015; Köhler et al., 2017).

The posterior anomaly of the mean DIC concentration in the deep ocean ($> 2000$ m) as the result of the 2500-year LGM integrations was in the range between 102 mmol/m$^3$ and 116 mmol/m$^3$, showing that the second guess DIC values that were used to initialize the respective LGM runs reasonably captured the target based on the estimate by Sarnthein et al. (2013).

The simulated sizes of the land carbon storage were similar among the three LGM runs and ranged between $1.44 \times 10^3$ GtC and $1.49 \times 10^3$ GtC. They were 340 GtC to 390 GtC smaller than in expPI (Table 3).

## 3.3 Carbon isotopes, phosphate, AOU, and the ideal age

The vertical structure of the distributions of biogeochemical tracers reflected the physical characteristics of each LGM experiment. The two experiments with a shallower AMOC (expLGMws and expLGMss) on the whole showed a similar structure

in the meridional sections of the Atlantic Ocean, while those for expLGM revealed markedly different features. For example, $\delta^{13}C_{DIC}$ values were higher in the upper ocean by $\sim1‰$ compared to those in expPI and lower in the deeper ocean to a similar degree in expLGMws and expLGMss (Fig. 5b,c), while this contrast was significantly smaller in expLGM (Fig. 5a). As a result, expLGMws and expLGMss demonstrated an excellent model-representation of $\delta^{13}C_{DIC}$, especially in terms of the depth that separates the positive and negative anomalies when compared to the data by previous studies (Peterson et al., 2014; Yu et al., 2020). The clear shallow–deep contrast of $\delta^{13}C_{DIC}$ anomalies was also in line with the estimated distribution by Oppo et al. (2018). In the Pacific Ocean, the negative anomalies indicated by data were well reproduced by all the LGM experiments except in the northern hemisphere in expLGMws (Fig. 5d-f). It should be noted that $\delta^{13}C_{DIC}$ was less equilibrated because of the uniform initial condition and that the drift of $\delta^{13}C_{DIC}$ in the last 500 model years in the deep North Pacific (at 30°N, 150°W, 2900 m) was 0.06‰ or less depending on the experiments, which was less than the typical magnitude of data uncertainty (e.g. Kurahashi-Nakamura et al., 2017).

A notable difference in the vertical structure in the Atlantic Ocean was also found in the distributions of dissolved phosphate (Fig. 6). In expLGMss and expLGMws, the anomaly in phosphate concentration reached more than 1 mmol/m$^3$ in the lower half of the depth range of the Atlantic Ocean, while it was negligible for the expLGM counterpart (Fig. 6a-c). The clear shallow–deep contrast of phosphate anomalies in expLGMss and expLGMws was in agreement with the inversion-based estimate by Oppo et al. (2018). These characteristics of phosphate distribution in the Atlantic Ocean were in accordance with the distribution of the AOU (Fig. 7a-c), which suggested that the strong positive anomalies of phosphate concentration in expLGMss and expLGMws were largely caused by an increase in remineralized phosphate. The distributions of both tracers showed patterns similar to each other also in the Pacific Ocean (Fig. 6d-f and 7d-f), similarly indicating the increase in phosphate concentrations due to amplified remineralization. Moreover, the inferred larger input of biological matter into the deep ocean was in accordance with the negative anomaly of $\delta^{13}C_{DIC}$ that covered most parts of the Pacific section. On the contrary, the parts that had smaller positive (or negative) anomalies of AOU corresponded to the positive anomalies of $\delta^{13}C_{DIC}$. Combined with the younger ideal age of seawater in the same regions (Fig. 8d-f), these results suggested that smaller storage of remineralized matter brought the positive anomalies of $\delta^{13}C_{DIC}$, which would have caused the clear mismatch between the simulated $\delta^{13}C_{DIC}$ in expLGMws and the data in the North Pacific.

The fact that the increased phosphate concentration in the deep water was mainly caused by an increase in remineralized phosphate was also supported by the higher fraction of remineralized phosphate in the total phosphate in the LGM experiments than in expPI (not shown). At the depths of 2000–3500 m in the Atlantic Ocean, expLGMws and expLGMss had older ideal age than expPI (Fig. 8b,c), suggesting that a more stagnant deep water resulted in the longer-lasting storage of remineralized organic matter, hence the increased phosphate. On the other hand, at the greater depths in the Atlantic and in most parts of the Pacific section below 2000 m, the effect of the increased soft-tissue pump in the mid-to-high latitudes of the southern hemisphere (Fig. 10a-c) and the subsequent northward transport of remineralized nutrients by the bottom circulation prevailed over the effect of less-efficient storage of nutrients by the younger water.

## 3.4 Carbonate ion concentration

The simulated carbonate ion concentration $[CO_3^{2-}]$ in all the experiments but expLGM showed a good agreement with the observation-based data in the Atlantic Ocean for the respective time periods especially at depths deeper than 2500 m, while the distribution suggested a penetration of the northern-sourced water in expLGM that was too deep (Fig. 9a-d). The increased
storage of remineralized carbon in the deep water for expLGMss and expLGMws resulted in lower $[CO_3^{2-}]$ accompanied by a lower pH of seawater as compared to those in expLGM, and then led to a larger vertical gradient of the concentration. However, the differences in the concentration between LGM and PI were larger than the reconstructed differences by Yu et al. (2020) (Fig. 9e-g). The effect of the increased storage of remineralized carbon was superposed onto the general rise of $[CO_3^{2-}]$ caused by the globally increased alkalinity, so that $[CO_3^{2-}]$ was higher in all the LGM experiments than in expPI in general in the
Atlantic Ocean and the Pacific Ocean as well. The elevated $[CO_3^{2-}]$ was not well in accord also with the estimated change in the deep Pacific that shows only little change in $[CO_3^{2-}]$ of no more than 5 mmol/m$^3$ (Yu et al., 2013), although the estimated increase in $[CO_3^{2-}]$ by ~25 mmol/m$^3$ in the Weddell Sea (Rickaby et al., 2010) was consistent with the results of expLGMws and expLGMss.

## 3.5 Export production

Although the global sum of the export production in the LGM experiments was similar to that in expPI (Table 3), the spatial distributions showed a remarkable difference between them (Fig. 10). On the one hand, in all the LGM experiments, the modeled carbon export exhibited a notable increase compared to the modern counterpart in the so-called high-nutrient, low-chlorophyll (HNLC) regions: the eastern equatorial Pacific and the mid-latitude band (40°S–60°S) in the southern hemisphere, in particular. The increased atmospheric dust input in the LGM experiments boosted the biological production in those regions by iron
fertilization (Fig. 10a-c), which was accompanied by more vigorous consumption of macronutrients resulting in significantly smaller concentrations of nitrate at the surface in those regions (Fig. 10d-f). The increase in carbon export was also seen in the upwelling region along the west coast of the African continent. This noticeable glacial increase in the carbon export in those regions was consistent with the reconstructed relative changes in export production provided by Kohfeld et al. (2005). There was no clear correlation between the distribution of carbon-export anomaly and the sea ice extent (Fig. 10a-c). Reduction of
the annual carbon export due to the expanded sea ice as discussed in previous studies (e.g. Kurahashi-Nakamura et al., 2007; Sun and Matsumoto, 2010; Gupta et al., 2020) was not significant in the southern hemisphere, which suggested that the dust effect of increasing the production overwhelmed the sea-ice effect. On the other hand, in the northern North Atlantic, the distributions of carbon export distinguished expLGMws from the other LGM runs. In expLGMws, the comparatively inactive deep convection in the northern North Atlantic caused a lower supply of macronutrients to the surface water, which resulted
in a smaller amount of carbon export (Fig. 10b) and a smaller nitrate concentration (Fig. 10e) in that region. In the other two LGM experiments, on the contrary, the more vigorous vertical mixing stimulated biological production leading to an increase in carbon export and less depleted nitrate concentrations.

### 3.6 CaCO₃ in the upper sediments

The global sum of the mass accumulation rate (MAR) of $CaCO_3$ for each experiment was 0.094 GtC/yr (expPI), 0.14 GtC/yr (expLGM), 0.12 GtC/yr (expLGMws), and 0.087 GtC/yr (expLGMss). Although the simulated modern MAR is ~25% smaller than the estimate (~0.13 GtC/yr) by Cartapanis et al. (2018), the LGM values except in expLGM approximated the estimate within the 2-sigma uncertainties for the glacial period (mean: ~0.11 GtC/yr, sigma: ~0.024 GtC/yr) by the same study.

The spatial distributions of the simulated $CaCO_3$ MAR also reasonably agreed with sedimentary data in the Atlantic Ocean, the Indian Ocean, and the Southern Ocean for all experiments (Fig. 11a-d), although the model underestimated the MAR in the eastern Pacific in expPI as in our preceding study (Kurahashi-Nakamura et al., 2020). The ratio of the glacial MAR to the modern value showed more distinguishable differences among the LGM experiments (Fig. 11e-g). In the North Atlantic, the reconstruction indicated a significantly lower MAR in the mid-latitude North Atlantic Ocean, which was only well reproduced in expLGMws. The anomalies of the $CaCO_3$ MAR in that region were governed by the supply of $CaCO_3$ to the sediments rather than its preservation and reflected a direct influence of the different magnitude of vertical mixing by local deep convection on the production in line with the results shown in Section 3.5. The export of $CaCO_3$ also depended on the sea-ice distribution (cf. Fig. 10a-c), because, in the BEC model, $CaCO_3$ production is scaled by the difference between local seawater temperature and the freezing point of seawater (Moore et al., 2004). In the South Atlantic, the MAR ratio was generally higher than the reconstruction due to the overestimated carbonate ion concentrations that provided more favorable environments for the preservation of $CaCO_3$. Therefore, expLGMss that had the least additional alkalinity showed the smallest ratio of MAR of the three LGM experiments. In the Southern Ocean, the simulated MAR of $CaCO_3$ was very low in all experiments, which resulted from low $CaCO_3$ productivity compared to opal fixation.

The spatial distributions of the $CaCO_3$ weight fraction in the upper sediment simulated by MEDUSA did not show distinctive differences among the LGM experiments (not shown). A key proxy-based criterion to assess the model results is the fact that the weight fraction for the LGM is lower in the Atlantic Ocean and higher in the Pacific Ocean than for the PI (Catubig et al., 1998), and all of the three LGM simulations satisfied this requirement despite the quite different global ocean states.

## 4 Discussion

### 4.1 Alkalinity and carbon inventories

A prerequisite for an acceptable marine biogeochemical state for the LGM is that the inventories of total alkalinity and DIC are compatible with the PI inventories. In other words, the differences in those ocean inventories between the two time periods need to be consistent with the changes in the inventories of the other reservoirs across the deglaciation. In the three LGM experiments of this study, 40 meq/m³ to 80 meq/m³ of alkalinity was added in addition to the increase due to the change in the seawater volume (Table 1). The excess of alkalinity corresponds to $0.5 \times 10^{17}$ eq to $1.0 \times 10^{17}$ eq in terms of the inventory in the ocean.

Assuming that those amounts of alkalinity were removed from the ocean by a net deposition of $CaCO_3$, $2.5 \times 10^{16}$ mol to $5.0 \times 10^{16}$ mol of $CaCO_3$ needed to be transferred to a different reservoir to account for the alkalinity decrease. Estimated amounts of the post-glacial shallow water deposition of $CaCO_3$ by coral reef buildup following the sea-level rise range from $2.2 \times 10^{16}$ to $18 \times 10^{16}$ mol (Milliman, 1993; Opdyke, 2000; Ridgwell et al., 2003; Vecsei and Berger, 2004; Husson et al.,

2018; Köhler and Munhoven, 2020), which would suggest that the extra alkalinity could be readily removed by that process. This is also supported by another estimation for the modern inorganic carbon burial in shallow water environments given by Cartapanis et al. (2018). The burial of $CaCO_3$ in deep-sea sediments in the pelagic oceans and the influx from the chemical weathering on land are other processes that affect the alkalinity inventory. Although the total alkalinity flux by these two processes during the past 20 kyr is uncertain, a simple estimate gives a net increase of alkalinity by $3 \times 10^{17}$ eq, if we assume

the input by land weathering is $3.5 \times 10^{16}$ eq/kyr and the removal by deep burial is $2 \times 10^{16}$ eq/kyr on average as discussed in Cartapanis et al. (2018). To counteract the net increase of alkalinity, $15 \times 10^{16}$ mol of $CaCO_3$ needs to be deposited, and the range of the estimated amount of $CaCO_3$ deposition by coral reef buildup covers the value as shown above (Fig. 12a).

Besides alkalinity, the compatibility with regard to the carbon inventory is examined as well (Fig. 12b). Out of the added $0.9 \times 10^{17}$ mol C to $1.2 \times 10^{17}$ mol C, which corresponds to 72 mmol/m$^3$ to 94 mmol/m$^3$, $2.5$–$5.0 \times 10^{16}$ mol was to be removed

by the shallow water deposition of $CaCO_3$. The rest ($4.0 \times 10^{16}$ mol to $9.5 \times 10^{16}$ mol, or 480 GtC to 1140 GtC), therefore, needs to be removed by other processes. The most probable one is the growth of the atmospheric reservoir. The increase in $pCO_2$ by 90 ppm between the LGM and PI corresponds to a 190 GtC expansion in the size of this reservoir. Another potential pool would be the terrestrial carbon reservoir. The modeled terrestrial reservoir in this study indicates a growth of 340 GtC to 390 GtC. The sum of the changes in both the reservoirs is 530–580 GtC, which leaves –100 GtC to 610 GtC (i.e. [480−580]

to [1140−530]) to be explained by further processes.

A plausible process to fill the gap could be again the post-glacial shallow-water deposition. An estimate of the modern deposition flux of organic carbon on shelves is 50–500 GtC/kyr (Cartapanis et al., 2018), which might suffice to remove the excess as long as it is a typical amount during a substantial portion of the Holocene, although there are no direct records of the organic-carbon deposition on continental margins over time (Cartapanis et al., 2018). Presumably, $CO_2$ release through pyrite

oxidation on exposed continental shelves during a glacial sea-level lowstand (Kölling et al., 2019) would add to the complexity of the change in carbon reservoirs across the deglaciation. It is expected that the sulfuric acid produced during pyrite oxidation is buffered with the dissolution of calcium carbonate minerals to lead to the release of $CO_2$. The associated $CO_2$ release rate is estimated to be between 12 GtC/kyr and 36 GtC/kyr if the acid is completely buffered by carbonate-mineral dissolution. The $CO_2$ release triggered by the erosion and oxidation of pyrite that happened underground during the LGM might be delayed

due to the transfer of $CO_2$ to the surface. For the LGM, a model-based estimate of the cumulative $CO_2$ release by buffering acids was 140 GtC (Kölling et al., 2019). As an extreme case, even if all of the released $CO_2$ reached the atmosphere during the deglaciation with a delay, that volume might be readily counteracted by the organic carbon deposition referred to above (50–500 GtC/kyr; Cartapanis et al., 2018). Another highly uncertain aspect is the relative change of the terrestrial-reservoir size. Various previous studies compiled in Kemppinen et al. (2019) and Jeltsch-Thömmes et al. (2019) suggest that the growth

of the terrestrial carbon reservoir from LGM to PI spans –500 to 1500 GtC, although the majority supports a (positive) growth

toward PI. The residual carbon excess (i.e. 480–1140 GtC) that needs to be removed apart from the contribution of the $CaCO_3$ deposition is in the range of these additional processes. To sum up, the removal of the whole excess of carbon is consistent with a combined contribution of the mechanisms mentioned above.

## 4.2 Simulated glacial biogeochemical states

Past studies attempted to constrain the LGM ocean state by utilizing the distribution of the stable carbon isotope ratio $\delta^{13}C_{DIC}$ (e.g., Tagliabue et al., 2009; Menviel et al., 2017; Kurahashi-Nakamura et al., 2017; Muglia et al., 2018; Gu et al., 2020; Kobayashi et al., 2021; Muglia and Schmittner, 2021). It turned out that this tracer is useful in determining the spatial distribution of water masses but much less in terms of the strength of the AMOC. Our study supports this conclusion. The noticeable negative anomaly of $\delta^{13}C_{DIC}$ in the deep ocean (deeper than $\sim$2500 m) found in the observation-based reconstructions (Pe-

terson et al., 2014; Oppo et al., 2018; Yu et al., 2020) was reproduced only in the case of a shallower AMOC, no matter what the strength of the AMOC is (expLGMss and expLGMws), which suggests that an ocean state having a shallower AMOC and northern-source deep water would be much preferable. A similar contrast between these shallower-AMOC LGM states and the deeper-AMOC LGM (expLGM) state is visible in the phosphate distribution, and again the shallower states show a much better correspondence to the reconstruction by Oppo et al. (2018), backing up the argument based on $\delta^{13}C_{DIC}$.

Another difference between the shallower and deeper AMOC states is found in the $[CO_3^{2-}]$ fields in the North Atlantic. The modeled distribution of $[CO_3^{2-}]$ in the shallower-AMOC experiments (expLGMss and expLGMws) indicates a vertical gradient between the depths of 1000 m and 4000 m that is noticeably larger than in expPI: larger by $\sim$50 mmol/m$^3$ in expLGMss, and by $\sim$70 mmol/m$^3$ in expLGMws, while only by $\sim$10 mmol/m$^3$ in expLGM. The reconstruction by Yu et al. (2020) shows a $\sim$50-mmol/m$^3$ larger vertical gradient for the LGM in the North Atlantic, which again supports the modeled states with a

shallower AMOC. An increase of the vertical gradient by a similar magnitude is also reported by Chalk et al. (2019). The increased vertical contrast in expLGMss and expLGMws is brought about mainly by the larger storage of carbon in the deep water due to a more sluggish circulation (2000-3500 m) and the larger amount of carbon export in the southern hemisphere. Contrary to the vertical gradient, however, the LGM experiments in this study overestimate the values of $[CO_3^{2-}]$ themselves in the North Atlantic by 20–50 mmol/m$^3$ compared to Yu et al. (2020). This would be, at least partly, due to the uniformly

increased alkalinity because such a systematic bias does not appear clearly in expPI. In the Southern Ocean, the same dataset by Yu et al. (2020) shows a slightly higher $[CO_3^{2-}]$ (by 10–20 mmol/m$^3$) in the bottom water for the LGM than for the modern, also supporting the results of expLGMss and expLGMws. On the other hand, the negative anomaly of $[CO_3^{2-}]$ in the South Atlantic caused by the expanded Glacial Pacific Deep Water proposed by Yu et al. (2020) is not reproduced in the experiments in our study.

Although the amounts of additional alkalinity applied in this study are in good agreement with previous estimates from the mass-balance point of view (see Section 4.1), the simulated $[CO_3^{2-}]$ is systematically too high most likely due to the uniformly increased alkalinity. This issue has two aspects. First, to manage the compatibility of the 190 ppm and more reasonable carbonate ion concentrations, one needs to realize the low $pCO_2$ with a smaller amount of added alkalinity. Supplementary LGM simulations without additional alkalinity resulted in higher $pCO_2$ by 20 to 43 ppm depending on the simulations compared to

the respective main simulations. The additional simulations depict ocean states that satisfy the constraint of deep-ocean carbon reservoir but lack the contribution of alkalinity increase. Although the contribution of additional alkalinity to $pCO_2$ drawdown seems to be minor compared to the whole change in $pCO_2$ since the LGM, as suggested by Ridgwell et al. (2003), other mechanisms would be required to achieve 190 ppm without (or, with a smaller amount of) additional alkalinity. For example, higher

5  solubility given by lower SST, a larger vertical contrast of DIC concentration by an even more stratified ocean (e.g. Kobayashi et al., 2021), and/or larger carbon storage in the deep water by a stronger biological pump (e.g. Morée et al., 2021). The more efficient carbon storage given by these processes would relax the problem of too-high carbonate ion concentrations. Second, the compatibility with the post-glacial shallow water deposition of $CaCO_3$ would need to be satisfied, too. In expLGMss that needed the smallest amount of added alkalinity of the three LGM experiments, the applied alkalinity corresponded to $2.5 \times 10^{16}$

10  mol of $CaCO_3$, which is already close to the lower limit of the independently-estimated amounts of the shallow water deposition (i.e. $2.2 \times 10^{16}$ mol). Therefore, to incorporate the likely post-glacial deposition of $CaCO_3$ and accompanying reduction of alkalinity into the evolution of the climate from the LGM to the modern, another source of alkalinity might be at work. More dissolution of $CaCO_3$ in the deep-ocean sediments or more input of alkalinity from the land weathering would be able to serve as the alkalinity source as discussed in Section 4.1. Future studies need to deal with the temporal variations of the global

$CaCO_3$ budget since the LGM, which is required for a more accurate discussion of the mass-balance during last 21 kyr.

This study utilized the properties of upper sediments with regard to $CaCO_3$ burial as a further criterion to distinguish among different physical ocean states. The preservation or dissolution of $CaCO_3$ in ocean-floor sediments is strongly affected by the carbonate chemistry of the surrounding seawater. Therefore, the degree of $CaCO_3$ preservation that is observed in the sediments is expected to contain local information on the properties of the seawater above the sediments. Although in this context

the solid weight fraction of $CaCO_3$ is a typical measure of the model–data fit, interpretation is not always straightforward because it is also affected by the amount of other solid species in the sediment. For example, even if the amount of $CaCO_3$ does not change, the increased dust input during the LGM would reduce the corresponding $CaCO_3$ weight fraction by "diluting" the contribution of $CaCO_3$. On the contrary, the MAR would not be affected by other solid components. In expLGMws, the $CaCO_3$ MAR in the North Atlantic Ocean (40–65°N) is lower than in expPI (Fig. 11) although the higher $[CO_3^{2-}]$ in the bot-

tom water is an advantage for the preservation of $CaCO_3$ in the upper sediments (Fig. 9). This fact suggests that the negative anomaly in the $CaCO_3$ MAR in that region is controlled by the supply of $CaCO_3$ to the sediments rather than its preservation, and it reflects a direct influence of the different (i.e. weaker) magnitude of vertical mixing by local deep convection leading to a lower supply of nutrients to the surface. Although the strength of deep convection is not directly related to the intensity of the meridional transport, a weaker convection may suggest a lower sea-surface density in the North Atlantic Ocean in some way,

and coincide with a reduced rate of meridional transport, considering the positive correlation between the AMOC strength and the meridional density gradient across the Atlantic Ocean (e.g., Rahmstorf, 1996).

In the LGM simulations of this study, the global carbon export by biological production is broadly similar to that in expPI (Table 3), which is in agreement with the estimates by previous model-based studies (Bopp et al., 2003; Tagliabue et al., 2009; Oka et al., 2011; Schmittner and Somes, 2016). The regional characteristic features of the modeled LGM carbon export (Sec-

tion 3.3) are well supported by other proxy-based studies. The increased nitrate consumption and carbon export in the Southern

Ocean are consistent with the estimate of glacial nutrient consumption based on nitrogen isotope ratios (e.g., Martínez-García et al., 2014; Kohfeld and Chase, 2017; Wang et al., 2017). Moreover, the decreased productivity in the northern North Atlantic Ocean obtained in expLGMws is in accordance with a reconstruction based on the analysis of dinocyst assemblages (Radi and de Vernal, 2008). This reconstruction suggests an annual productivity lower by 50–150 gC/m$^2$ during the LGM, which is reproduced reasonably well in the experiment. As discussed in the previous paragraph, the reduced productivity is also consistent with the MAR data. The change in the distribution of biological production naturally influences the distribution of phosphate and AOU. The distinctive increase of carbon export in the Southern Ocean brings larger concentrations of both tracers in the bottom water in spite of its younger ideal age compared to expPI, which contributes to the good model–data agreement with the reconstructed phosphate distribution in the Atlantic Ocean. A similar feature is found in the very deep water in the Pacific section, while the increase in the export production in the northern hemisphere together with the older ideal age produces very high anomalies of phosphate and AOU in the depths of 500–1500 m of the North Pacific Ocean.

The younger ideal age in the Southern Ocean of the LGM experiments is caused by the changes in local convective mixing rather than by the changes in AABW flow. expLGM has a very similar magnitude and geometry of AABW to that in expPI (Fig. 3), but nevertheless the ideal age in the Southern Ocean is significantly younger. On the other hand, in expLGMws and expLGMss, more vigorous AABW does not convey the comparatively old water at the depths of 2000–3500 m in the Atlantic to the south of $\sim$40$^\circ$S. In addition, the LGM experiments have a deeper mixed layer depth in the Southern Ocean, which would contribute to the better ventilated Southern Ocean.

## 4.3   Relationship between the DIC concentrations and water-mass ages

The estimates of the DIC inventories in the deep ocean during the LGM that we used in this study for model calibration are based on the assumption that there is a linear relationship between the local water age and DIC concentrations in the modern deep ocean below the depth of 2000 m that is also applicable to the glacial ocean (Sarnthein et al., 2013). The same assumption is employed in another study for estimating the size of the deep-ocean carbon pool (Skinner et al., 2015). We simulated the carbon age by adding reservoir ages estimated for the modern surface water in different basins (Matsumoto, 2007) and the estimated increase of the surface age in the LGM (Skinner et al., 2017) to the explicitly modeled ideal age, and examined the DIC–age relationship in the model states (Fig. 13). An analogous quasi-linear relationship also appears in our model results from the pre-industrial run (Fig. 13a). It turns out, however, that the LGM model oceans have a different structure of the DIC–age relationship (Fig. 13b-d): in the Pacific and Southern Oceans the slope of the linear regression lines is less steep and the intercept is substantially larger, while in the Atlantic Ocean it clearly has a steeper slope.

The reasons for the different structures are twofold. First, the remarkably different slopes of the regression lines for the Atlantic Ocean and the other two oceans are caused by the DIC enrichment of the southern-sourced deep water due to the reinforcement of the biological soft-tissue pump by the increased dust input to the high-latitude Southern Ocean. The carbon-rich southern-sourced deep water strongly influences the Southern and Pacific Oceans, so that these two oceans are generally high in DIC compared to the modern deep water. On the other hand, the properties of the Atlantic water can be explained by

the mixing of northern-sourced water and the DIC-enriched southern-sourced water, leading to the apparent steeper gradient for the Atlantic Ocean.

The separation of the Atlantic branch from the others (Fig. 13b-d) becomes invisible by confining the depth domain to deeper ranges in some LGM runs, so that an overall linear relationship with a relatively similar slope as in the modern case is obtained. The mixing effect becomes invisible in similar scatter plots but for water deeper than 3000 m for the LGM simulations with a shallower AMOC (Fig. 13g,h), while the northern-sourced component still emerges in the LGM run with a deeper AMOC (Fig. 13f) because the northern-sourced water reaches the depth of 3000 m. In the pre-industrial setting, a very similar relationship is still valid also for the water deeper than 3000 m (Fig. 13e).

Besides the influence of the enriched southern-sourced water, as a second factor, the prior addition of mean DIC to the entire ocean in the LGM simulations would raise the regression lines uniformly. Considering a possible alteration of the DIC–age correlation induced by these two factors, applying the modern regression line to the glacial condition does not necessarily provide an accurate estimate of the glacial mean DIC concentration in the deep ocean.

In the instances of this study, the DIC concentration according to the modern regression line (Fig. 13a) and those based on the glacial lines (Fig. 13b-d) are different approximately by $\sim$100 mmol/m$^3$ when the mean age of deep water is 2500 years. More than half of the difference is caused by the modified mean DIC concentration in the entire ocean (see Table 1), and the rest results from the carbon enrichment due to the increased biological pump. Unfortunately, it is not straightforward to estimate a more accurate DIC–age relationship for the LGM in the framework of this study because the magnitude of the two influencing factors, especially the mean DIC offset, is uncertain. We adjusted the mean total alkalinity so that it was compatible with the prescribed mean DIC and the 190 ppm in the atmosphere. Although this means that the mean DIC concentration, in turn, can be tuned to yield the 190 ppm with a prescribed alkalinity inventory, it would be also difficult to have a reliable estimate of the alkalinity inventory during the LGM due to the large uncertainty of the observation-based estimate of the alkalinity budget by the post-glacial $CaCO_3$ deposition (see Section 4.1).

## 5 Conclusions and outlook

Three time-slice simulations for the LGM with a comprehensive Earth-system model including a global carbon cycle module were carried out, so that reasonable biogeochemical states in terms of the fit to paleoclimatological and paleoceanographic records have been achieved by tuning of the total inventories of DIC and alkalinity in the ocean. The simulated ocean states are expected to serve as an initial state for future transient simulations of the last deglaciation, i.e. from the glacial (LGM) to the interglacial (PI) climate state. In terms of biogeochemical tracer distributions in the ocean, our results clearly show that the LGM ocean states with a shallower AMOC are characterized by tracer distributions that can be more easily reconciled with the relevant reconstructions as suggested by various previous studies. Model–data comparisons regarding $CaCO_3$ properties in the upper sediments add further support to a state with a weaker and shallower AMOC rather than a stronger circulation, although they do not constrain directly the magnitude of the volume transport.

The examination of the balance of the bulk ALK and DIC budgets shows that all the LGM simulations in this study are likely to be compatible with the PI state within the uncertainties of the available constraints, at least from a mass balance point of view. The question whether they would indeed reach the PI state following a realistic trajectory would need to be examined in a transient context, because the trajectory would depend on the timing and magnitude of the deglacial sea-level rise that governs the post-glacial deposition on the shelves. Considering that the sea-level rise is a direct consequence of the ice-sheet evolution induced by climate changes, it would be fundamental to analyze and discuss the evolution of the coupled carbon cycle–ice sheet system.

As a connection between these two subsystems, shelf or shallow-water processes including the weathering and deposition of biogeochemical matter and their modeling (e.g. Munhoven and François, 1996; Kölling et al., 2019; Börker et al., 2020; Lacroix et al., 2020) would take an important role. Those processes would be fundamental to the evolution of the global carbon cycle at the glacial–interglacial timescale not only for the post-glacial evolution, but also during the evolution in the time period that preceded the LGM. The excess alkalinity was prescribed and added to the ocean in the LGM experiments of this study to satisfy independent observation-based constraints, but in reality, it would be determined as the cumulative imbalance between incoming and outgoing fluxes during the course of the evolution from the last interglacial period to the LGM, which will be another target of transient simulations in the future.

*Code and data availability.*  The newly developed model source codes to tailor CESM1.2 for experiments with additional freshwater forcing and model output of the main experiments will be available at https://doi.org/10.1594/PANGAEA.942327.

*Author contributions.*  TK-N developed the model code for the additional freshwater forcing with input from UM. TK-N and AP designed the experiments, and TK-N carried them out. TK-N interpreted and discussed the results with contributions from all co-authors. AP, UM, and MS conceptualized the overarching research project. TK-N prepared the manuscript with contributions from all co-authors.

*Competing interests.*  This study has no competing interests.

*Acknowledgements.*  The authors would like to thank the reviewers for their insightful comments and suggestions. This research was funded by the project PalMod (www.palmod.de; FKZ: 01LP1505D) within the framework of Research for Sustainable Development (FONA, http://fona.de) by the German Federal Ministry for Education and Research (BMBF).

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

**Table 1.** Overview of the experimental designs in this study

| experiments | general setting | FW forcing in N-ATL[1] (Sv) | FW forcing in SO[1] (Sv) | prior addition of mean DIC (mmol/m$^3$) | prior addition of mean ALK (meq/m$^3$) |
|---|---|---|---|---|---|
| expPI | DECK[2] | – | – | – | – |
| expLGM | PMIP4 | – | – | 94 | 80 |
| expLGMws | PMIP4 | +0.1 | – | 72 | 60 |
| expLGMss | PMIP4 | −0.1 | −0.25 | 75 | 40 |

1: see also Fig. 1

2: Eyring et al. (2016)

3: Kageyama et al. (2017)

**Table 2.** The first-guess adjustment of the global mean concentrations of DIC and total alkalinity for the preparatory runs, and the resultant DIC anomaly averaged over the depths of more than 2000 m. In the last column, the differences between each of the LGM runs and expPI that are averaged over the last 50 years are shown.

| experiments | prior addition of average DIC (mmol/m$^3$) | prior addition of average ALK (mmol/m$^3$) | posterior DIC anomaly below 2000 m (mmol/m$^3$) |
|---|---|---|---|
| prepLGM | 100 | 88 | 106 |
| prepLGMws | 100 | 100 | 128 |
| prepLGMss | 100 | 75 | 125 |

**Table 3.** Biogeochemical properties of the model states (the last 50-years average) in this study. For DIC, the anomaly is globally averaged over the depths of more than 2000 m and shows the difference between each of the LGM simulations and expPI.

| experiments | $p$CO$_2$ | DIC anomaly below 2000 m | export production[1] | land C storage |
|---|---|---|---|---|
| | (ppm) | (mmol/m$^3$) | (GtC yr$^{-1}$) | (GtC) |
| expPI | 276 | – | 7.6 | $1.83\times10^3$ |
| expLGM | 185 | 109 | 7.8 | $1.49\times10^3$ |
| expLGMws | 186 | 102 | 7.4 | $1.44\times10^3$ |
| expLGMss | 182 | 116 | 7.3 | $1.48\times10^3$ |

1: global export production at 100-m depth including POC and $CaCO_3$

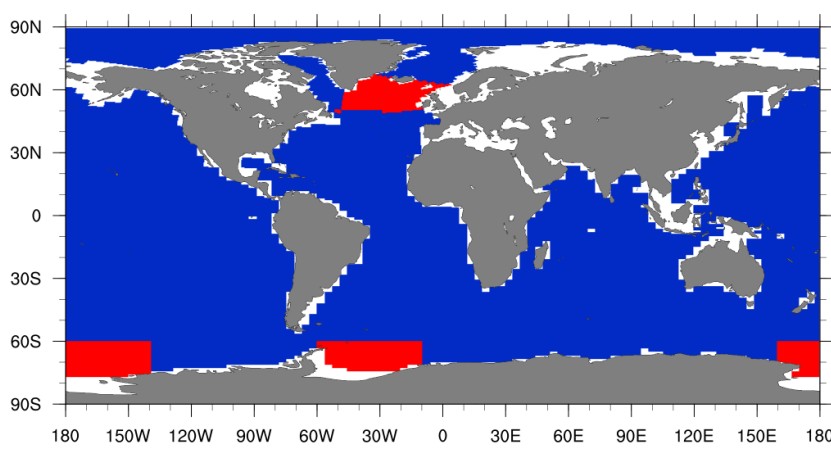

**Figure 1.** Regions allotted for the additional freshwater forcing (shown in red). Additional 0.1 Sv in total was uniformly given to the specified region in the North Atlantic for expLGMws, while, for expLGMss, 0.1 Sv and 0.25 Sv were subtracted from the North Atlantic and Southern Ocean, respectively. Corresponding compensation (i.e. freshwater flux having the opposite sign) was applied to the other regions homogeneously to keep the total volume of sea water constant.

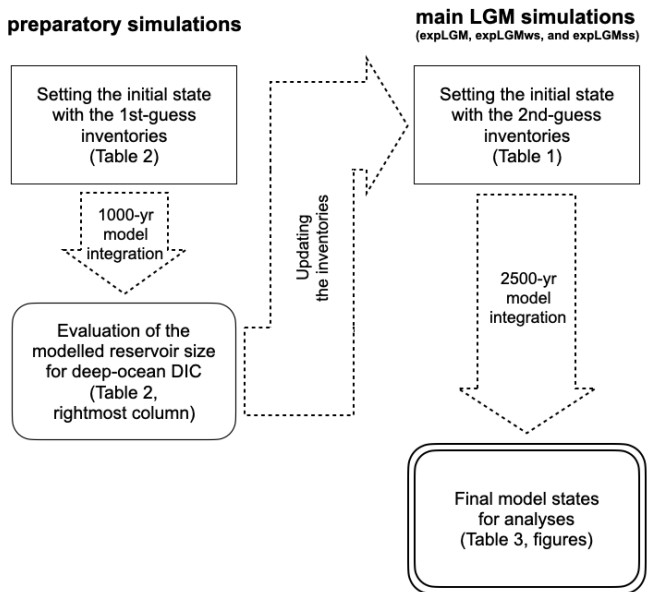

**Figure 2.** Workflow for the LGM carbon-cycle simulations in this study.

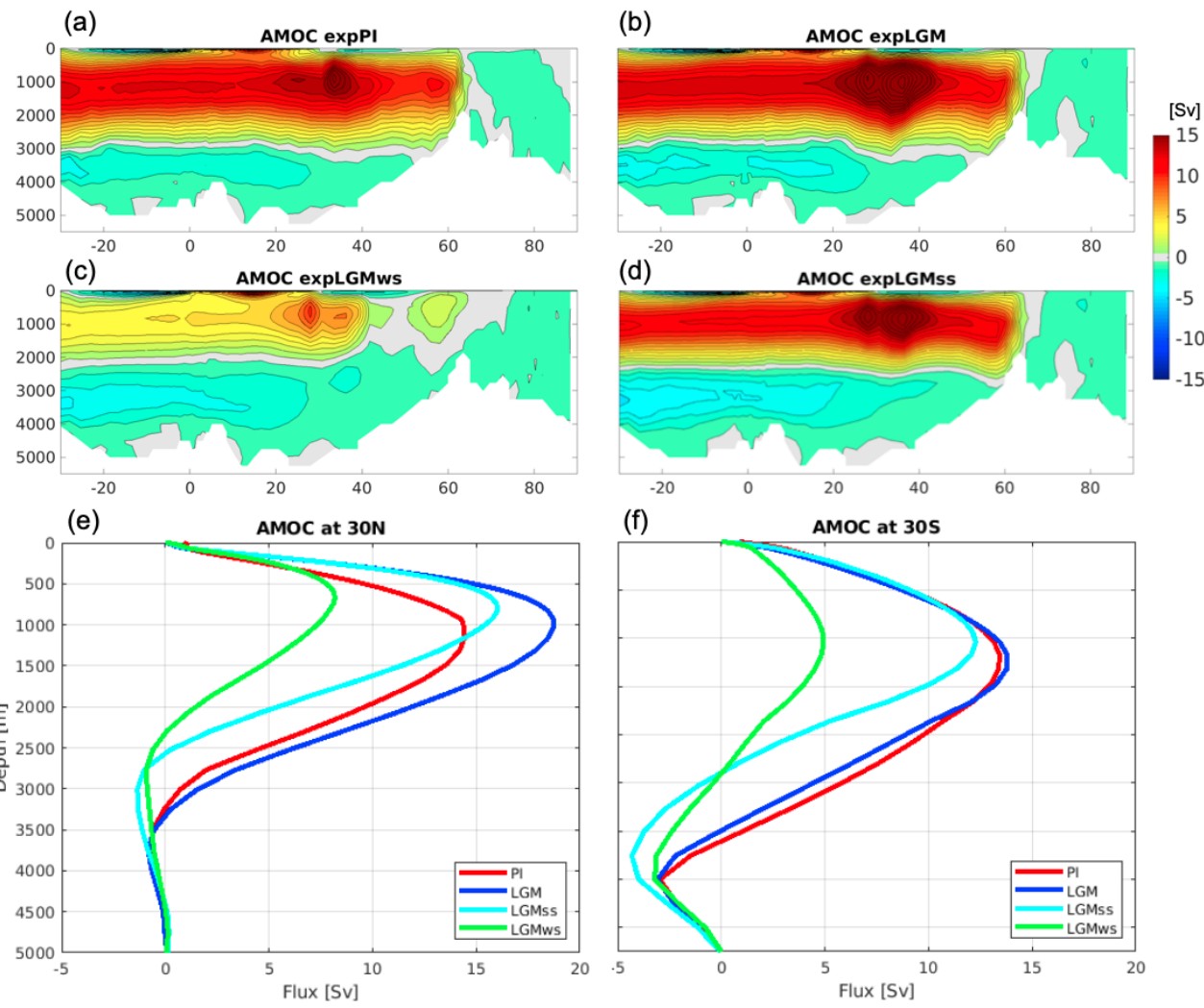

**Figure 3.** Simulated AMOC for the main experiments in this study (Sv): stream function for (a) expPI, (b) expLGM, (c) expLGMws, and (d) expLGMss. The vertical profiles of the AMOC (e) at 30N and (f) at 30S are also shown. The average states of the last 50 years are shown.

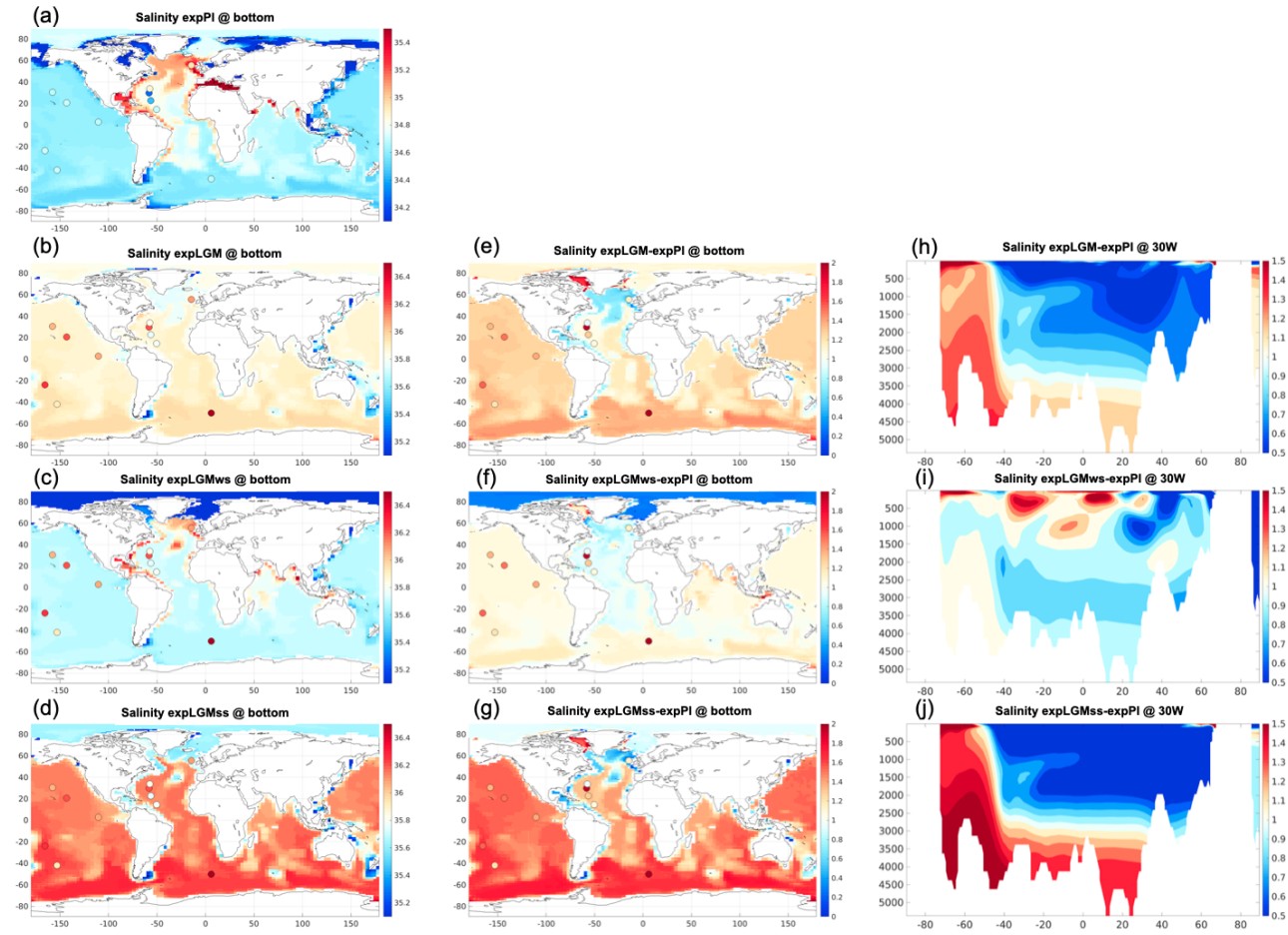

**Figure 4.** Simulated salinity for the main experiments in this study. The left and middle column show the global distributions of salinity in the bottom-most grid cells. The absolute values for (a) expPI, (b) expLGM, (c) expLGMws, and (d) expLGMss, and the differences between each LGM experiment and expPI are shown (e-g). The colored dots indicate the reconstructions by Adkins et al. (2002), Insua et al. (2014) and Homola et al. (2021). The meridional sections in the Atlantic (30W) are also shown for the anomalies (h-j). The average states of the last 50 years are shown.

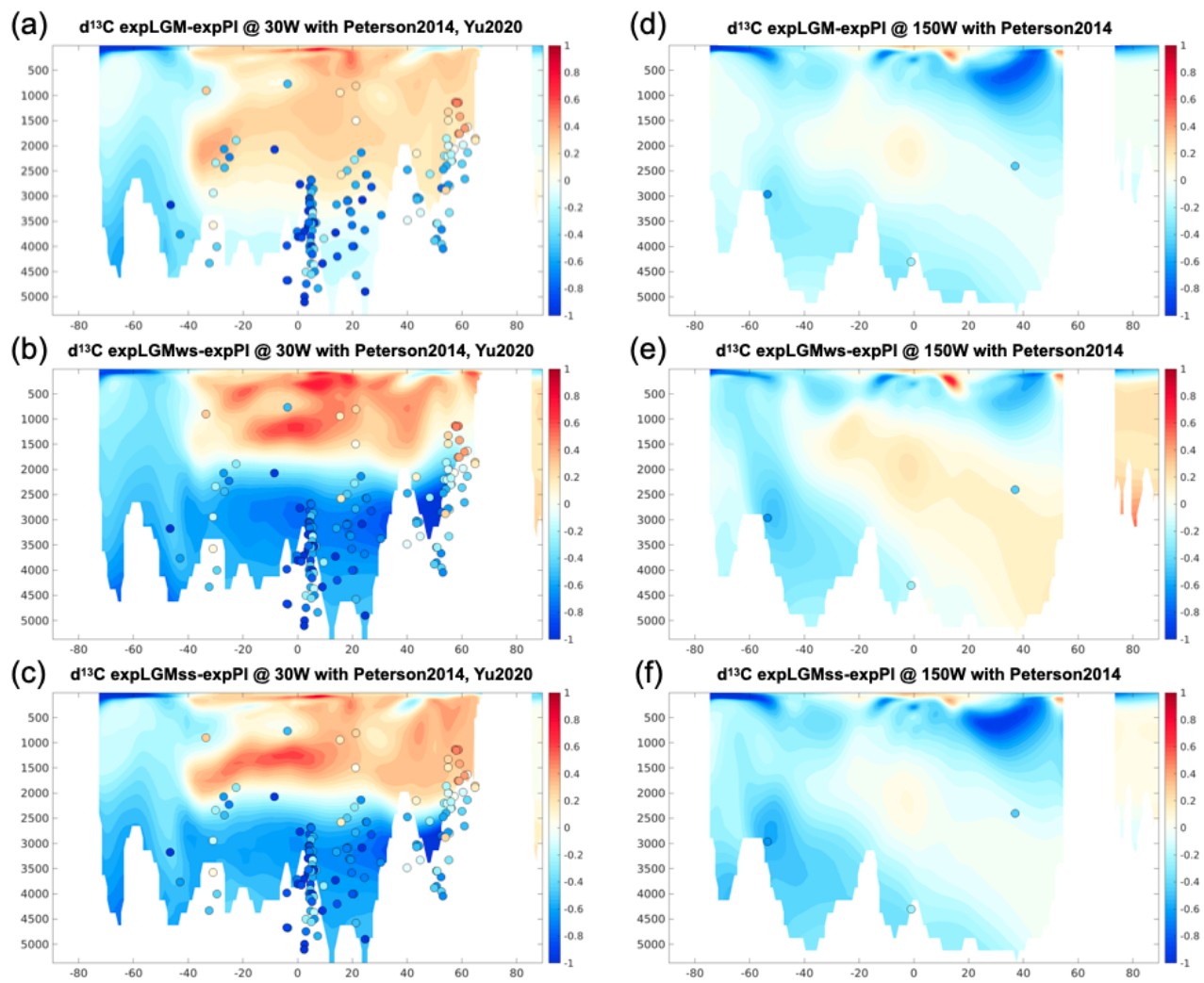

**Figure 5.** Meridional sections in the Atlantic (a-c; at 30W) and in the Pacific (d-f; at 150W) of the simulated $\delta^{13}C$ of DIC in the LGM experiments (‰): expLGM (a,d), expLGMws (b,e), and expLGMss (c,f). The differences between each LGM experiment and expPI are shown. The average states of the last 50 years are shown. The dots indicate the reconstructions by Peterson et al. (2014) and Yu et al. (2020).

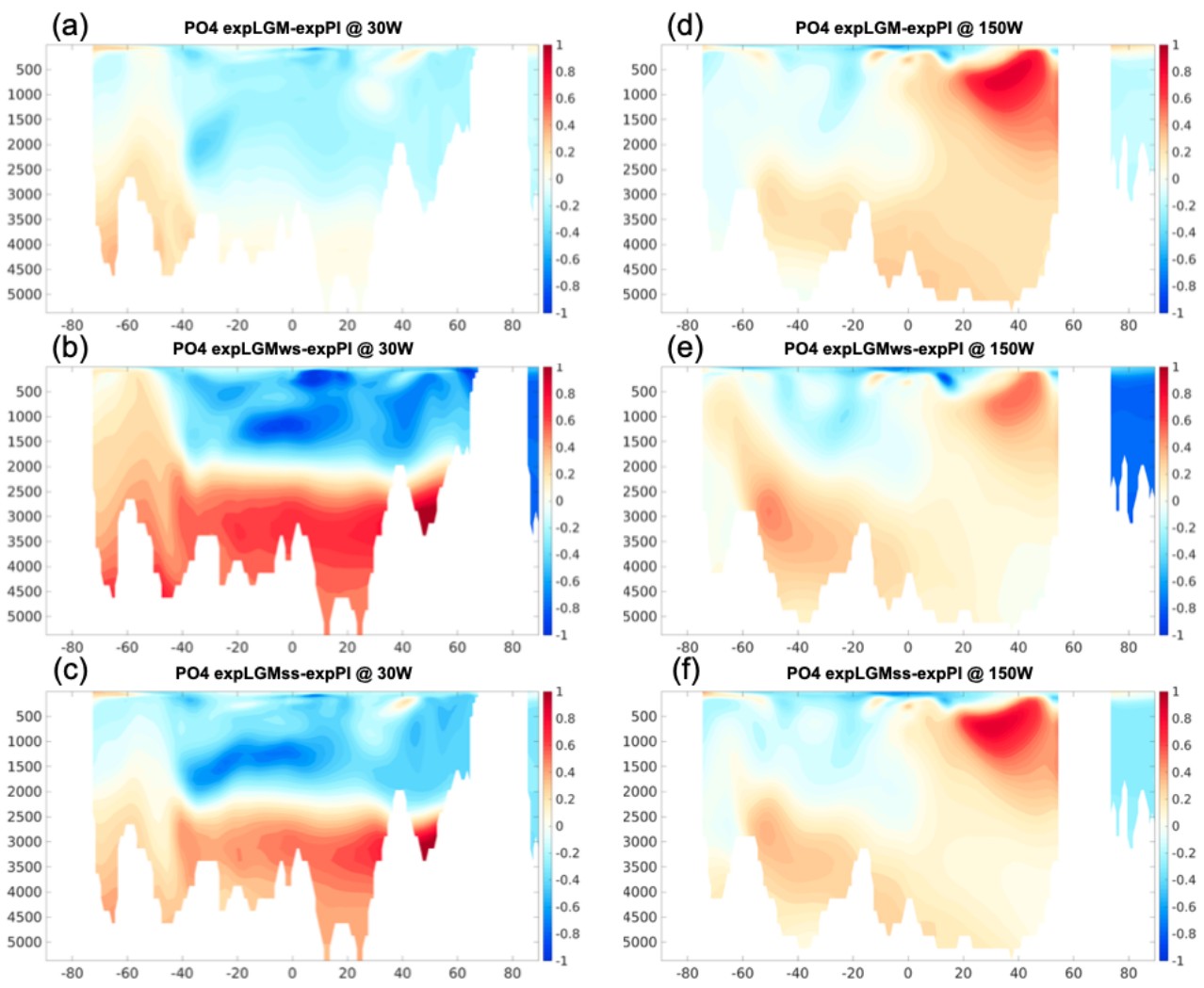

**Figure 6.** Meridional sections in the Atlantic (a-c; at 30W) and in the Pacific (d-f; at 150W) of the simulated phosphate concentrations in the LGM experiments (mmol/m$^3$): expLGM (a,d), expLGMws (b,e), and expLGMss (c,f). The differences between each LGM experiment and expPI are shown. The average states of the last 50 years are shown.

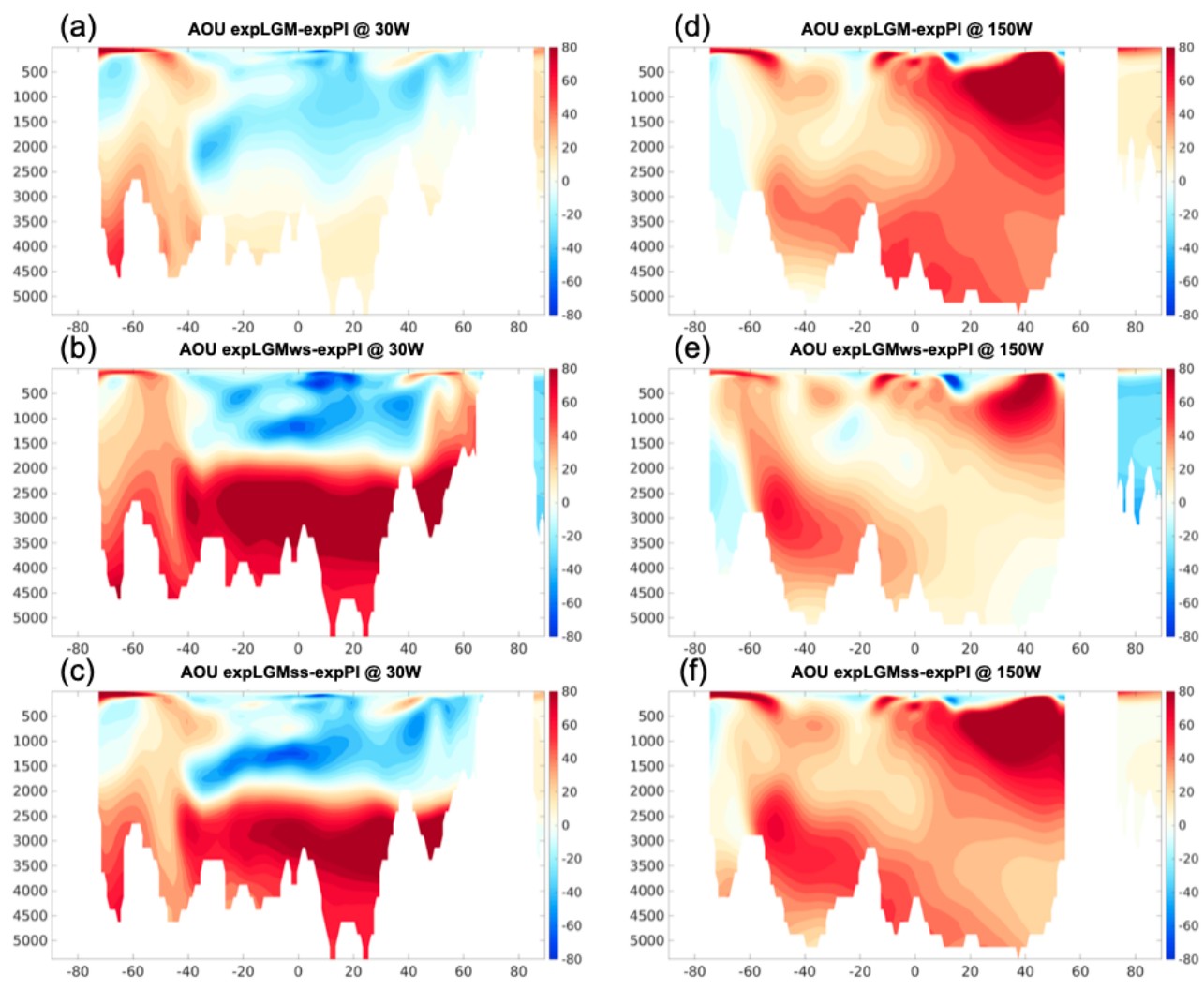

**Figure 7.** Meridional sections in the Atlantic (a-c; at 30W) and in the Pacific (d-f; at 150W) of the simulated AOU in the LGM experiments (mmol/m$^3$): expLGM (a,d), expLGMws (b,e), and expLGMss (c,f). The differences between each LGM experiment and expPI are shown. The average states of the last 50 years are shown.

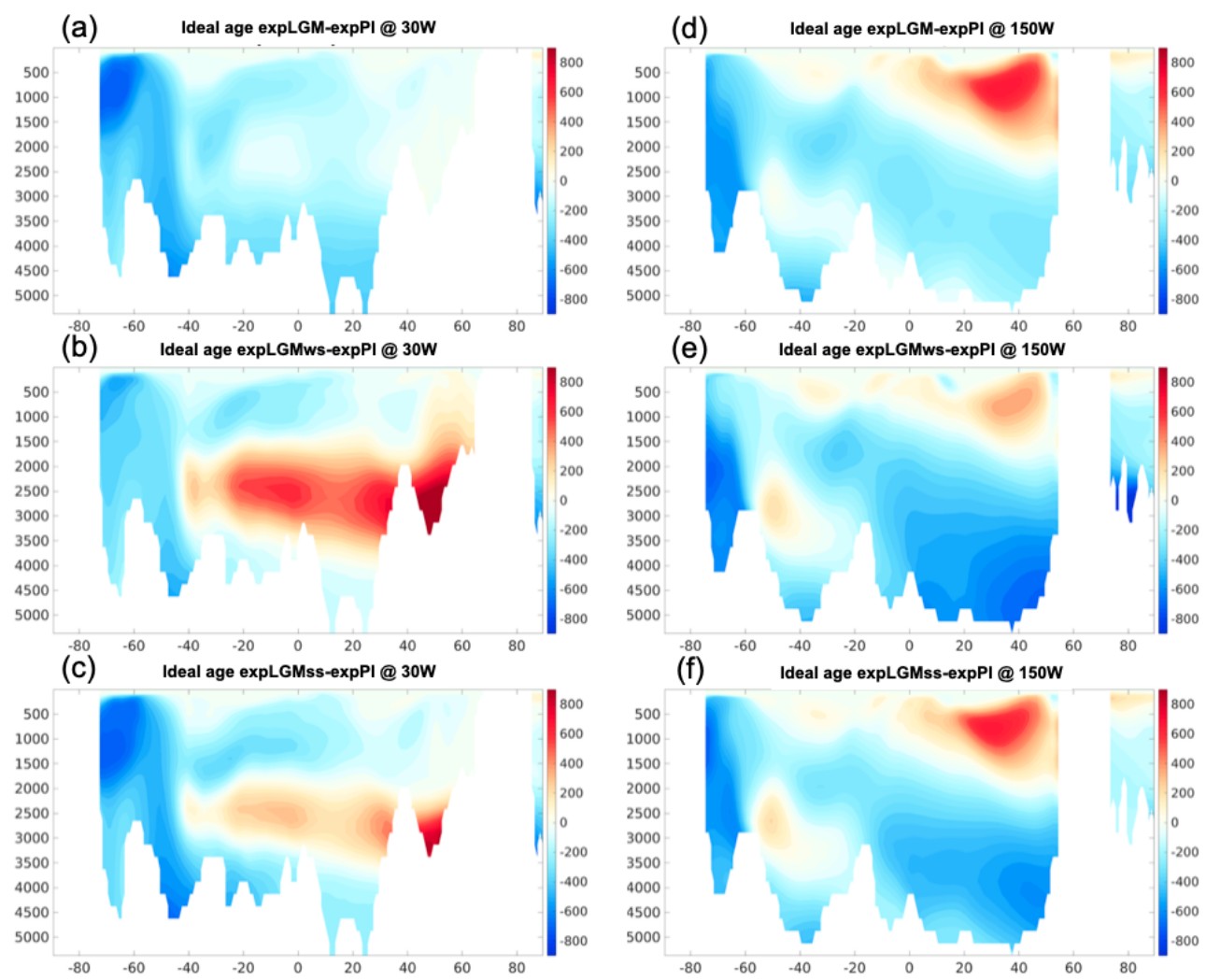

**Figure 8.** Meridional sections in the Atlantic (a-c; at 30W) and in the Pacific (d-f; at 150W) of the simulated ideal age of sea water in the LGM experiments (years): expLGM (a,d), expLGMws (b,e), and expLGMss (c,f). The differences between each LGM experiment and expPI are shown. The average states of the last 50 years are shown.

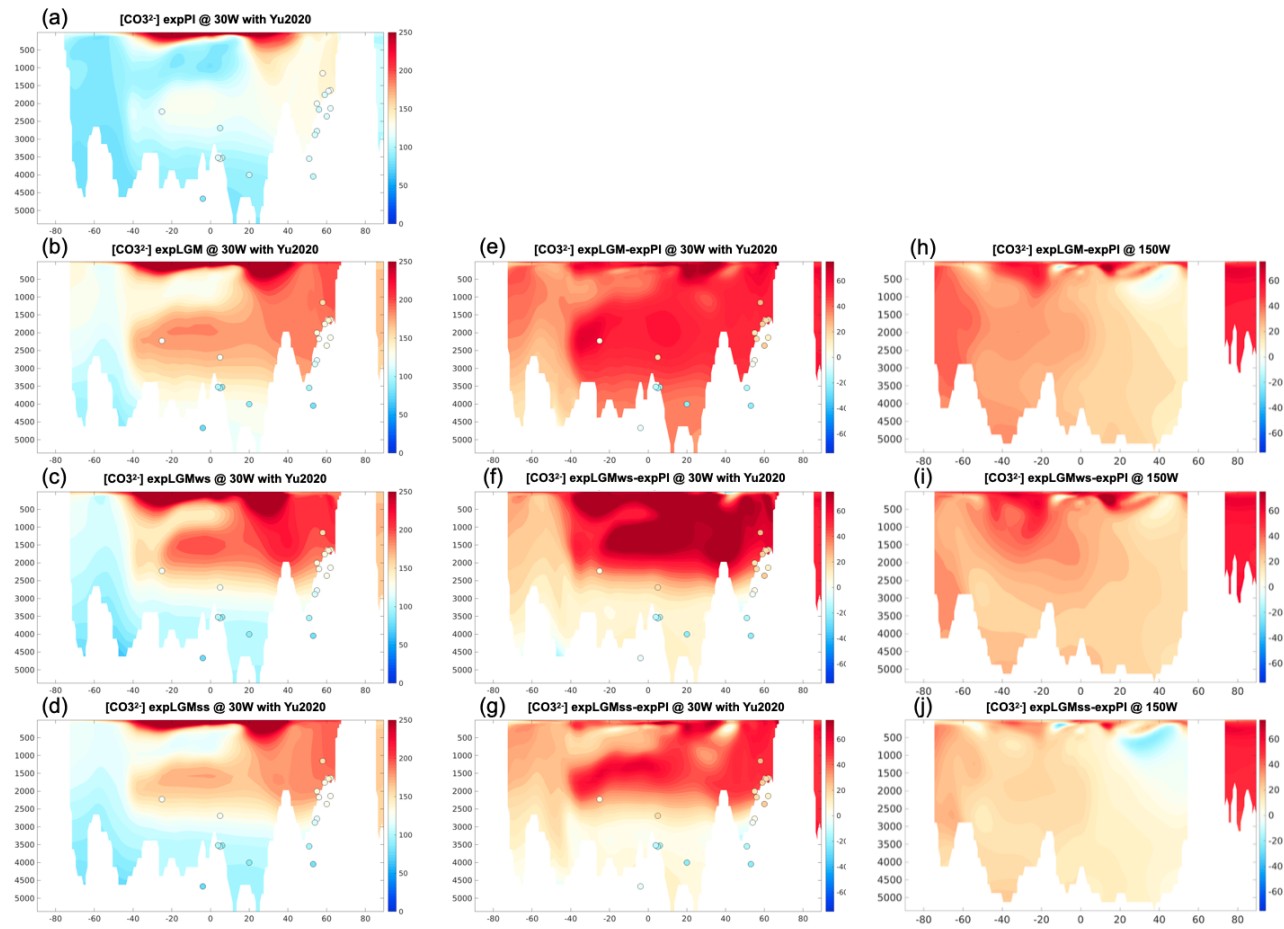

**Figure 9.** Meridional sections in the Atlantic (a-g; at 30W) and in the Pacific (h-j; at 150W) of the simulated carbonate ion concentrations (mmol/m³). The absolute values for (a) expPI, (b) expLGM, (c) expLGMws, and (d) expLGMss, and the differences between each LGM experiment and expPI are shown (e-j). The dots indicate the reconstruction by Yu et al. (2020). The average states of the last 50 years are shown.

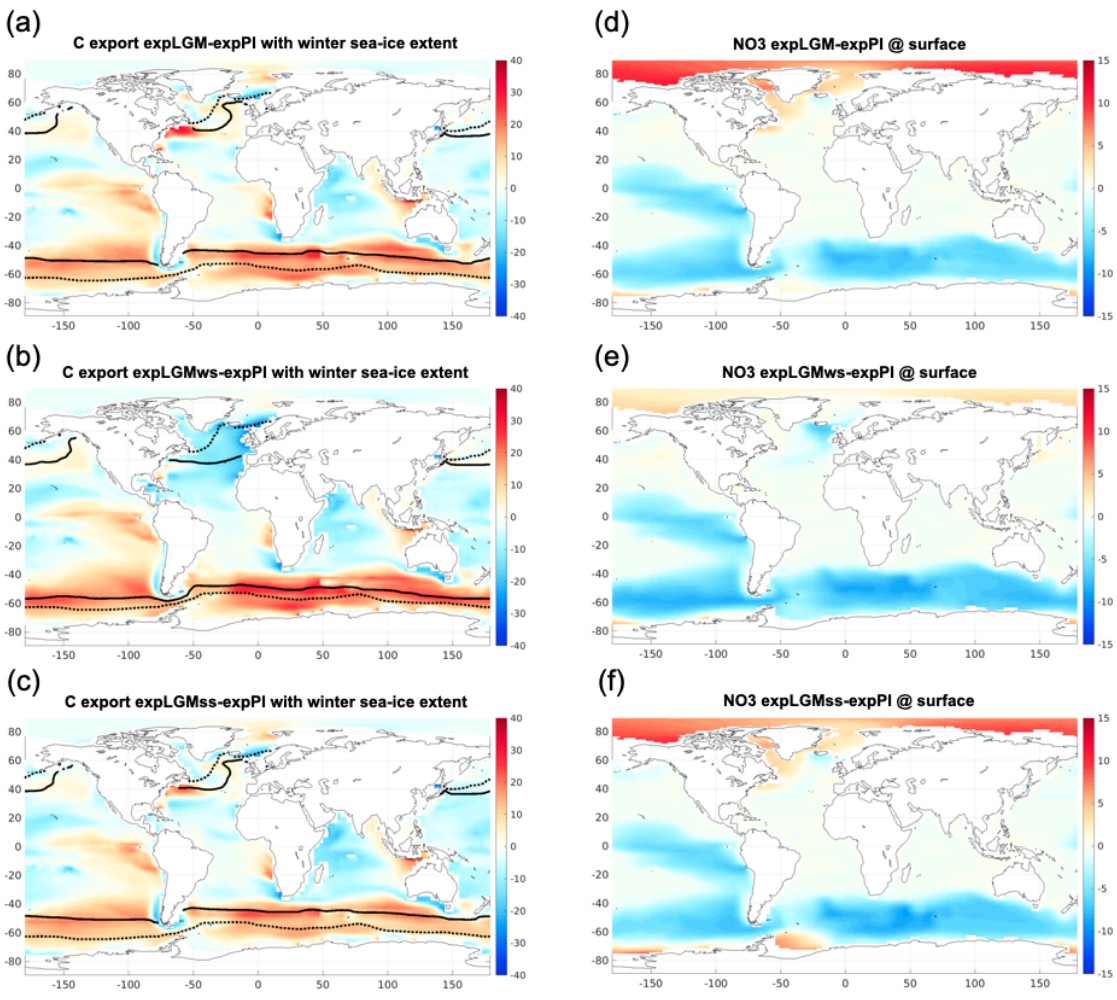

**Figure 10.** Carbon export at a depth of 100 m (left panels; gC/m$^2$/year) and the concentration of nitrate at the surface (right panels; mmol/m$^3$). The differences from those in expPI are shown for each of the LGM runs: expLGM (a, d), expLGMws (b, e), and expLGMss (c, f). In the left panels, the maximum extent of sea ice during the winter season of the respective hemisphere is also shown as solid lines (LGM) and dotted lines (PI). The average states in the last 50 years are shown.

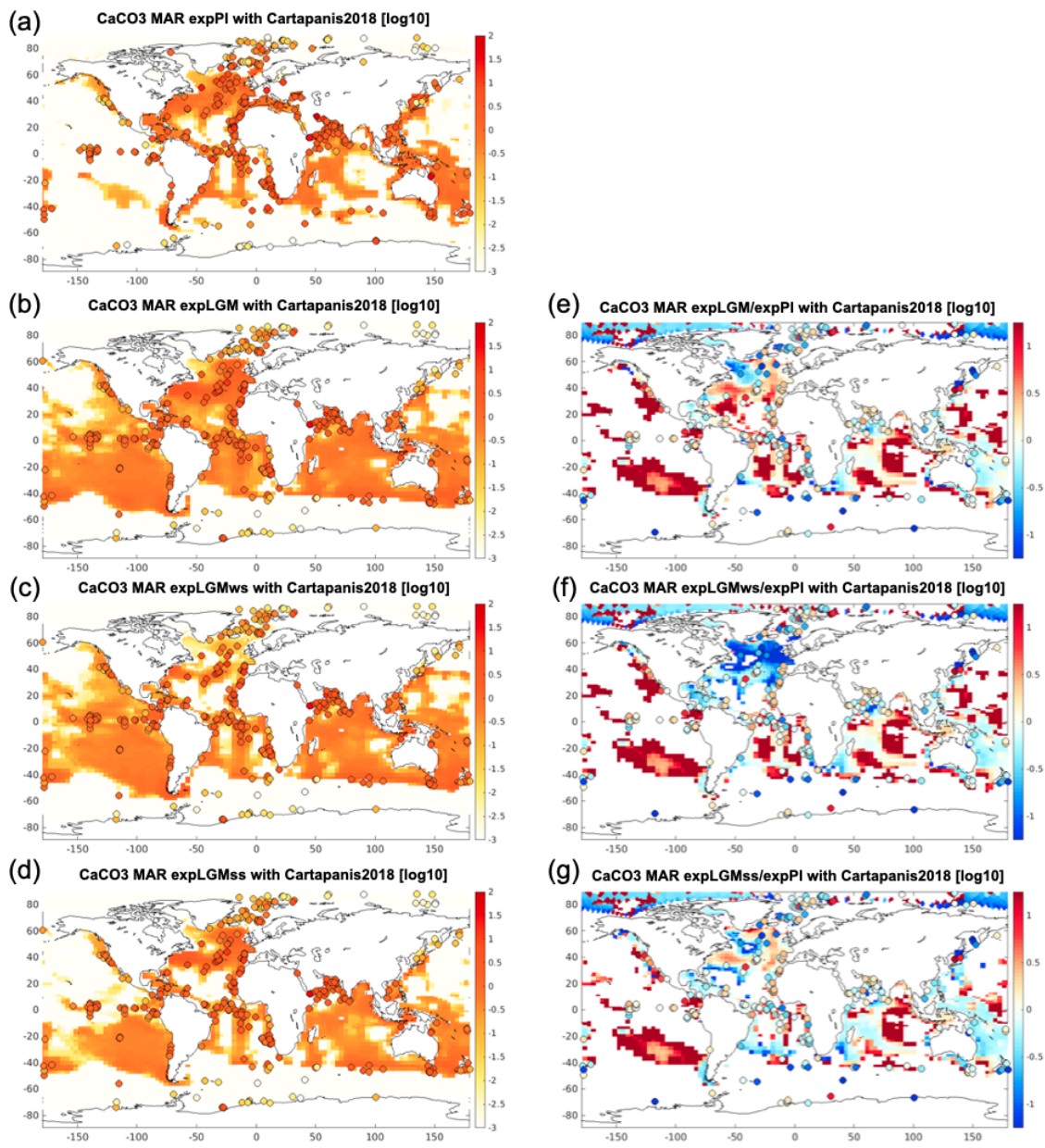

**Figure 11.** Mass accumulation rate (MAR) of $CaCO_3$ in the upper sediment simulated with MEDUSA. The left column shows the absolute values (g/cm$^2$/kyr) for (a) expPI, (b) expLGM, (c) expLGMws, and (d) expLGMss on a logarithmic scale with overlaid dots indicating the observation-based data by Cartapanis et al. (2018). The ratio of MAR in the respective LGM runs to that in expPI run is shown in the right panels: (e) expLGM, (f) expLGMws, and (g) expLGMss. For the ratio, the locations with MAR less than $1 \times 10^{-10}$ g/cm$^2$/ky were removed from the plot domain to highlight regions of large contrasts.

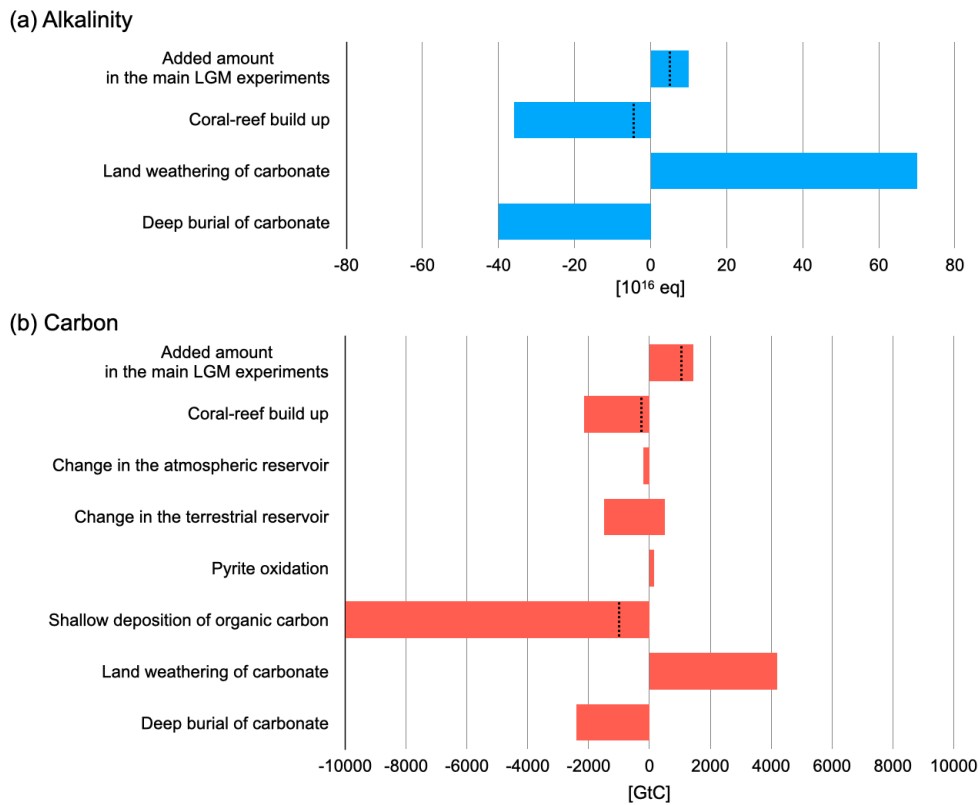

**Figure 12.** Processes discussed in Section 4.1 that contributed to the change in the ocean inventories of alkalinity (a) and carbon (b) since the LGM. Positive values indicate addition of alkalinity or carbon to the ocean and negative values removal of them. For the component "coral-reef build up", the range of estimates by previous studies (Milliman, 1993; Opdyke, 2000; Ridgwell et al., 2003; Vecsei and Berger, 2004; Husson et al., 2018; Köhler and Munhoven, 2020) is shown. "Land weathering of carbonate", "deep burial of carbonate" and "shallow deposition of organic carbon" are based on the time-average fluxes estimated by Cartapanis et al. (2018). "Change in the atmospheric reservoir", "change in the terrestrial reservoir" and "pyrite oxidation" show the amount from ice-core records (Bereiter et al., 2015; Köhler et al., 2017), the range of estimates by previous studies (Kemppinen et al., 2019) and the estimate by Kölling et al. (2019), respectively. The dotted lines correspond to the lowest estimate for the respective components.

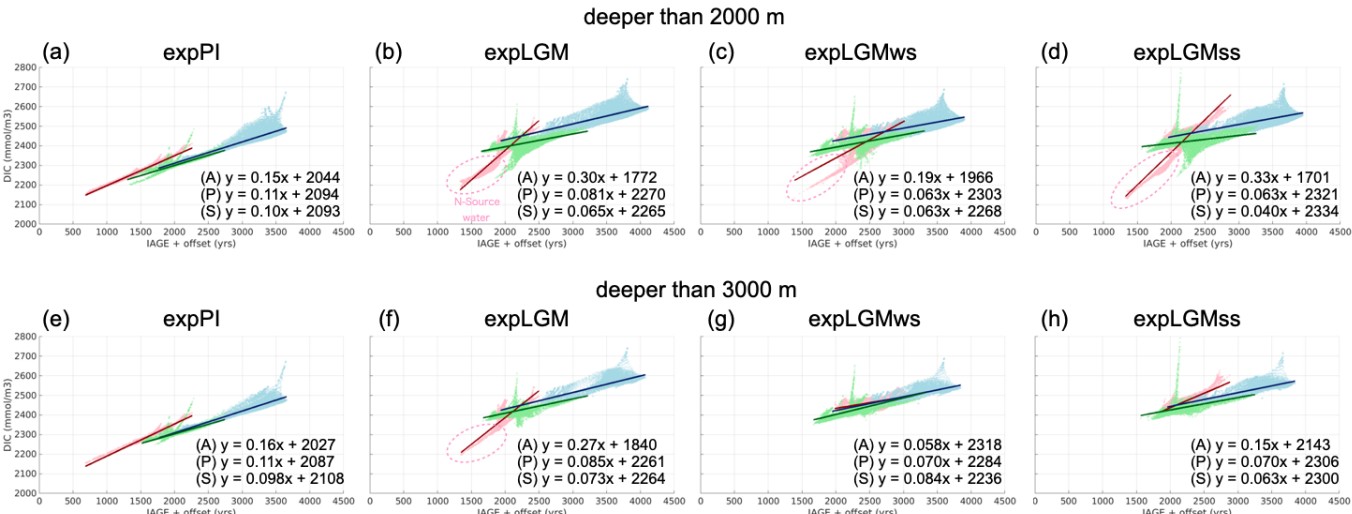

**Figure 13.** Scatter plots showing the correlations between DIC concentrations and the age of local water for the Atlantic (red), the Pacific (blue), and the Southern Ocean (green). The age was calculated by adding estimated reservoir ages for the surface water (Matsumoto, 2007; Skinner et al., 2017) to the modeled ideal age. The linear-regression coefficients are also shown at the lower-right of each panel, where $x$ denotes the simulated age (years) and $y$ the concentration of DIC (mmol/m$^3$). The upper (lower) four panels show the properties of water that is deeper than 2000 m (3000 m).