# Peer review of "Glacial state of the global carbon cycle: time-slice simulations for the last glacial maximum with an Earth-system model"

_Climate of the Past, 2022_

## Author Comment (AC1)

[Figure]

**Figure C1.** Global maps of the difference in $p$CO$_2$ between the surface ocean and the atmosphere ($\Delta p$CO$_2$) for the three LGM experiments in this study (a-c). Where $\Delta p$CO$_2$ is positive, $p$CO$_2$ is higher in the surface ocean, corresponding to a net source for atmospheric CO$_2$. The differences in $\Delta p$CO$_2$ between each LGM simulation and expPI are also shown (d-f).

---

## Author Comment (AC2)

[Reply to Reviewer #1]

We appreciate the constructive and beneficial comments by the reviewer.

Q1: In the introduction, the authors emphasized the importance of transiency to explain the glacial-interglacial carbon cycle. I would like to see more discussion of the time scale of the response of the carbon cycle. How fast do carbonate sediments change in response to changes in sea level? How do the time scales of carbonate sediment expansion and contraction differ from other processes that consist of the ocean carbon cycle (solubility, biological pumps, and ocean circulation)?

A1: We will re-organize the introduction to a great extent, and will include the description about the timescale of various processes relevant to the carbon cycle. We will thereby make the concept of this study clearer.

Q2: Although the authors focus on shallow-water carbonate sediments, carbonate is thought to be equally buried in the deep ocean (Cartapanis et al., 2018). I would like to see a clearer separation of these two contributions in the discussion.

A2: This comment by the reviewer is totally reasonable. We will include relevant descriptions in a reorganized introduction, and will extend the discussion in Section 4.1 by including deep-ocean processes. As the reviewer pointed out, in the mass balance calculation to connect the glacial and modern states, the outflow of CaCO3 through deep-sea burial over the entire periods of time since the LGM should be also taken into account. At the same time, we will also mention a possible importance of the inflow by the land weathering, because, depending on the magnitudes of the outflow and inflow, they would have worked as another source or sink of alkalinity as a whole.

Q3: In the LGM experiments, whole ocean alkalinity is increased to adjust the atmospheric CO2 concentration to the ice core data. The magnitude of this increase does not seem to be explicitly stated (not shown in Table 2), but is it appropriate?

A3: The magnitude of alkalinity adjustment is already described in the right-most column of Table 1, and appears also in the first paragraph of Section 4.1. See also A6 below.

Q4: In section 4.1, the authors explained the change in alkalinity is related to the changes in shallow-water coral reefs but is it necessary to consider the effect of carbonate compensation, including deep-sea carbonate sediments (e.g., Brovkin et al., 2012; Ganopolski and Brovkin, 2017; Kobayashi et al., 2021)?

A4: As the reviewer pointed out, processes related to the deep-sea carbonate sediments should be included in the discussion about the compatibility of alkalinity inventory between LGM and PI in Section 4.1. As answered in A2, we will extend the section to include the discussion about the flux of alkalinity by deep-sea carbonate burial and that by the land weathering as well based on the quantitative estimates for each of them (rather than the mechanism of carbonate compensation itself) because the imbalance between them would matter in the context of the discussion.

Q5: It is reported that PMIP models tend to simulate lower ocean carbon sequestration and higher atmospheric CO2 if they had a lower ocean volume at the LGM (Lhardy et al., 2021). Do the ocean bathymetry and volume change in this study? It is expected that a greater change in alkalinity would be required if a lower ocean volume is adopted.

A5: Yes, we changed the bathymetry for the LGM, and the effects of sea-water volume change are included in the simulations. As described in Section 2.2, we applied the volume-change effects following the PMIP4 protocol (Kageyama et al., 2017). The volume change alters the concentration of alkalinity and other tracers, but does not affect the inventories.

Specific comments:

Q6: P1/L6: "The increase..." I am not convinced by this statement because of the lack of information on the inventory change in alkalinity.

A6: This question is closely related to Q3. Although relevant information is already available as we answered in A3, we will also specify the required increase of alkalinity inventory in the main text of a revised method section.

Q7: P2/L15: How much carbon (PgC) does the change in DIC in the deep ocean (µmol/kg) between the LGM and the present day correspond to?

A7: It is reported that the changes in concentration correspond to 730–980 PgC (Sarnthein et al., 2013) and 687 PgC (Skinner et al., 2015) in the respective papers. We will add those numbers in PgC (or GtC) to the introduction.

Q8: P2/L16: I would like to know more about the coral reef hypothesis and the subsequent study's discussion of the impact of changes in shallow-water carbonate sedimentation on global carbon cycle changes.

A8: As answered in A1, we will reorganize the introduction and will add more description of the coral reef hypothesis.

Q9: P2/L20: I would like to know the explicit statement about the impact of changes in DIC and alkalinity on atmospheric CO2 concentrations. In other words, the increase in DIC in the deep ocean during glacial periods and the increase in alkalinity throughout the ocean both contribute to a decrease in atmospheric CO2.

A9: The increase in the deep-water DIC storage should be the outcome of combined two factors: the change in the vertical structure of carbon storage and the change in the mean concentration. The increased vertical contrast contributes to the decrease in  $pCO_2$  as the increase of alkalinity does, while the increase of mean DIC concentration has the opposite effect. The adjustment of DIC inventory in the entire ocean of this study will provide the net effect of the two factors that is consistent with the estimated deep DIC storage. We will add these description to the introduction.

Q10: P3/L21: Would you explain a little more about the advantages of using SolveSAPHE?

A10: SolveSAPHE is based on a more robust algorithm to provide better numerical stability than the original pH solver, while it can offer comparable accuracy of pH calculation. As far as we experienced, the original solver was frequently highly sensitive to the chemical composition of seawater, so that it often caused numerical instability especially in a spin-up phase. We therefore adopted SolveSAPHE and continued to use it for consistency.

Q11: P3/L30: Changes in carbonate sediment burial result in changes in whole ocean alkalinity (Kobayashi et al., 2021). Is it difficult to investigate the changes in carbonate burial by using the sediment model?

A11: The amount of global carbonate burial was not available in principle in the application of MEDUSA in the remapped (1deg x 1deg) domain, because it did not cover all the ocean floors of the POP2 domain. For a revised manuscript, we coupled MEDUSA simply or directly to every bottom grid cell of POP2 as in the preceding study (Kurahashi-Nakamura et al., 2020) to involve the whole ocean floors and carried out similar simulations again. This new method has enabled us to provide the global sum of MAR of CaCO3 for each experiment, which are 0.094 GtC/yr (expPI), 0.14 GtC/yr (expLGM), 0.12 GtC/yr (expLGMws), and 0.087 GtC/yr (expLGMss). Although the modeled modern MAR is ~25% smaller than the estimate (~0.13 GtC/yr) by Cartapanis et al. (2018), the LGM values approximate the +-2-sigma range of the estimate for the glacial period (the mean: ~0.11 GtC/yr) by the same study. Another advantage of the new method is that now we can provide global continuous maps of MAR. We will introduce new plots for CaCO3 MAR in a revised manuscript.

Q12: P3/L30: I would like to see information on carbonate burial fluxes and sediment distribution throughout the ocean as calculated by the sediment model in expPI. p7/L8 includes a brief description, but I do not think it is sufficient.

A12: As answered in A11, in a revised manuscript we will introduce a new coupling scheme between MEDUSA and CESM to cover the global ocean, so that we will be able to provide further MEDUSA-related results.

Q13: P4/L13: Would you cite a reference for the Ruddiman belt?

A13: Although here we show a relevant reference "W.F. Ruddiman, Late Quaternary deposition of ice-rafted sand in the subpolar North Atlantic (lat 40 to 65N), Geol. Soc. Am. Bull., 88 (1977), pp. 1813-1827", we will delete the words in a revised manuscript following a comment by the other reviewer.

Q14: P4/L15: Why did you choose 0.25 Sv for the freshwater input to the Southern Ocean?

A14: We did test runs in which we tried various different amounts of fresh water for the additional forcing, and empirically found 0.25 Sv to realize a shallower-but-stronger AMOC structure. That is to say, we needed to subtract that amount of freshwater for the southern-sourced water to counteract the strengthened northern-sourced water.

Q15: P4/L34: What is the total change in alkalinity adjusted for "second-guess"? Would you add the information to Table 1?

A15: The information is already available in the right-most column of Table 1, and appear also in the first paragraph of Section 4.1.

Q16: P5/L14: Would you consider showing the stream function as well?

A16: We will add stream-function plots as well in a revised manuscript.

Q17: P5/L19: From my understanding, LGMss was an experiment in which freshwater was removed from the LGM in the North Atlantic and fresh water was added in the Southern Ocean, resulting in a stronger north-south density gradient at the sea surface. However, why do we see a somewhat weaker AMOC relative to the LGM?

A17: We assume that this is a misunderstanding by the reviewer. Freshwater was removed also from the Southern Ocean as shown in the text and in Table 1.

Q18: P5/26: How about comparing the modeled deep-sea salinity to paleo records (e.g., Adkins et al., 2002; Insua et al., 2014; Homola et al., 2021?

A18: We will add new plots to show the comparison between the model results and the data given by the three studies.

Q19: P6/L9: Can we assume that this change is consistent with estimates of changes in terrestrial carbon storage (e.g., Peterson et al., 2014; Jeltsch-Thömmes et al., 2019)?

A19: Yes, as discussed in Section 4.1 by referring to Kemppinen et al. (2019), our model results show the growth of terrestrial carbon storage of 340--390 GtC, which is within the range of estimates by various previous studies including Peterson et al. (2014). Although Jeltsch-Thömmes et al. (2019) is not included in the compilation of previous work, their estimate (450 to 1250 GtC) is also within the range of the uncertainty, which does not affect our discussion. We will add the latter article to the reference in the revised manuscript.

Q20: P6/L9: Is the total amount of carbon in the atmosphere, ocean, and terrestrial reservoir the same for all experiments? It is difficult for me to understand the experimental design.

A20: No, it isn't. Each experiment has a different total amount of carbon stored in the entire (atmosphere-ocean-land) system. In our experiments, instead of fixing the total amount, we tuned the size of the atmospheric reservoir to a specific amount (i.e. ~190 ppm), which practically governed the terrestrial reservoir's size, and also tuned the size of the deep-ocean reservoir as well. The size of the shallow-ocean reservoir varied among different experiments depending on the vertical gradient of DIC concentration in the ocean, hence the ocean circulation. The difference in the total carbon inventory in the entire system can be therefore interpreted as the uncertainty of the total carbon inventory that arose from the uncertain ocean circulation. By adding a similar description, we will modify the manuscript to better explain the experimental design.

Q21: P6/L14: Would you show us d13C paleo records (e.g., Peterson et al., 2014) in the figure? It would make comparisons with other studies easier.

A21: We will introduce new figures of the modelled  $\delta^{13}C_{DIC}$  with overlaid plots of paleo records by Peterson et al. (2014) and Yu et al. (2020).

Q22: P6/L22: The impact of changes in the distribution of export production on nutrients and AOUs should also be considered. How about conducting sensitivity experiments with fixed biological fluxes?

A22: We will add discussion about the topic by introducing new plots for PO4 and AOU. Although we find the sensitivity experiments suggested by the reviewer intriguing, we judged that they are beyond the scope of this particular work because the model configurations currently available do not allow us to do that and demand further substantial technical development.

Q23: P6/L25: The decrease in ideal age in the Southern Ocean in Fig. 4 is caused by changes in AABW flow or changes in local convective mixing. Please describe.

A23: The younger ideal age in the Southern Ocean of the LGM experiments is caused by the changes in local convective mixing rather than by the changes in AABW flow. expLGM has a very similar magnitude and geometry of AABW to that in expPI, but nevertheless the ideal age in the Southern Ocean is significantly younger. On the other hand, in expLGMws and expLGMss, more vigorous AABW does not convey the comparatively old water at the depths of 2000-3500 m in the Atlantic to the south of ~40S. In addition, the LGM experiments have a deeper mixed layer depth in the Souther Ocean, which would contribute to the better ventilated Southern Ocean. We will add a similar description to a revised manuscript.

Q24: P6/L26: Some studies have reconstructed carbonate ions from B/Ca (e.g., Rickaby et al., 2010; Yu et al., 2013, 2020). Would you compare your modeling results with them? As stated in the discussion, the increase in alkalinity seems to be overestimated in the current setting.

A24: We will update the plots for the carbonate ion concentrations with overlaid plots of the data by Yu et al. (2020) to facilitate the model-data comparison. We will also refer to the other two papers that have very sparse data points in a discussion section for further comparisons.

Q25: P6/L31: How about showing the reconstructed changes in export production (Kohfeld et al., 2005)? The characteristics of the changes appear to be well reproduced in the model. Also, I would like to see a discussion of the effects of sea ice distribution and iron fertilization on export production.

A25: We will refer to Kohfeld et al. (2005) to update the discussion about the modeled changes of the export production fields. We will add description and discussion to the main text because Kohfeld et al. (2005) only provides the qualitative changes. We will also update the plots for the export production with overlaid sea-ice extent in respective LGM experiments and will discuss the effects of sea ice distribution and iron fertilization on export production with some additional references such as Kurahashi-Nakamura et al. (2007), Sun and Matsumoto (2010), and Gupta et al. (2020).

Q26: P7/L12: Would it make sense to compare this model-data comparison of CaCO3 MAR for other ocean regions (the Southern Ocean and the Pacific Ocean)? It makes the model validity and shortcomings clearer.

A26: As mentioned in A11, we will introduce new plots for CaCO3 MAR in a revised manuscript to show the model-data comparison in the global ocean.

Q27: P8/L15: The pyrite oxidation showed here as negative feedback on atmospheric CO2, but was insufficiently studied to constrain its quantitative contribution to the glacial-scale carbon cycle. Is my understanding of this correct?

A27: Yes. The pyrite oxidation accompanied by  $CO_2$  release is expected to occur (at least get triggered) during the lowstands of sea-level changes under the glacial low- $pCO_2$  environments, and therefore would potentially work as a negative feedback in the  $pCO_2$  variations in that sense. However, to specify the actual role of the pyrite oxidation in the glacial-interglacial cycles, not only the amount of the associated  $CO_2$  release but also its timing need to be better constrained.

Q28: P8/L32: A recent modeling study of Kobayashi et al. (2021) also used d13C to constrain their ocean carbon cycle fields in the LGM.

A28: We will add the reference in a revised manuscript to the line specified.

Q29: P9/L32: I asked a similar question about the export production, but does the change in sea ice coverage affect the sinking flux of CaCO3 in this region?

A29: Unlike the total export production, the export of CaCO3 is sensitive to the sea-ice distribution, because, in the BEC model, CaCO3 production is scaled by the difference between local seawater temperature and the freezing point of seawater (Moore et al., 2004). We will add this description to a revised manuscript.

Q30: P10/L7 Typo? The references are not listed correctly.

A30: We will correct it.

Q31: P10/L13: The maximum values of ideal age in the Pacific appear to be getting younger in expPI, expLGM, and expLGMss (expLGMws). Is this related to the increased AABW-related deep-water flow?

A31: Yes, as the reviewer pointed out, the ideal age of Pacific water is younger in the LGM runs because the fresher southern-sourced deep water is more dominant. In revised Section 4.3, however, we will totally modify the sea-water age and relevant discussion by taking the reservoir age of surface water into account.

Reference:

[revised manuscript text omitted]

---

## Author Comment (AC3)

[Reply to Reviewer #2]

We appreciate the constructive and beneficial comments by the reviewer.

Major points

Q1: In the introduction the authors emphasize the importance of simulations covering the entire glacial cycle to also capture the effects of slow processes. While it is clear that this is currently not possible due to the prohibitive computational costs, it would be interesting to discuss what effects and impacts would be expected from these slow processes and whether they could bias the results of the present study.

A1: In a revised manuscript, we will reorganize the introduction to a large extent to include the description about the timescale of various processes. We will thereby make the concept of this study clearer.

Q2: While by design of the experiments the largest changes are expected to be in the Atlantic, there are surely also important differences in the physical ocean states and biogeochemistry of the rest of the ocean. However, this is neither discussed nor shown in any of the figures.

A2: We will add plots for a Pacific section for various tracers, and add discussion about them accordingly.

Q3: In the two shallow LGM runs (LGMsw and LGMss) large changes in phosphate and carbonate ion concentrations and AOU exist. The authors argue that this is either related to the more sluggish deep ocean ventilation or the biological carbon pump. Yet, the ideal age tracer distributions, that in fact indicate younger bottom water in the entire Atlantic, and the stronger stream function in the deep Atlantic strongly suggest that this was only caused by the more efficient biological carbon pump in the Southern Ocean. Due to the importance of this process and the far-reaching effects I would like to see a more in-depth discussion and analysis of this matter.

A3: We appreciate the beneficial suggestion. We should have discussed the depths of 2000-3500 m and the deeper (>3500 m) depths separately. In the former depth range, expLGMws and expLGMss had older ideal ages, which contributed to the more efficient storage of remineralized matter. However, in the latter depths, we agree with the reviewer that the effect of the increased biological pump in the mid-to-high latitudes of the Southern hemisphere and the northward transport of remineralized nutrients by the bottom circulation prevailed over the effect of the younger age of the corresponding water. We will add these description to a revised manuscript. (Please see also A26 and A27)

Q4: In section 4.1 the authors mention that the applied alkalinity changes are in good agreement with previous estimates for carbonate deposition during the deglaciation. However, in section 4.2 it is then mentioned that the [CaCO3] are systematically too high most likely due to the uniformly increased alkalinity. How can this be reconciled?

A4: This discussion would have two aspects. First, to manage the compatibility of the 190 ppm and more reasonable carbonate ion concentrations, one needs to realize the low $pCO_2$ with the smaller amount of appended alkalinity. This would require the help of other mechanisms to reduce $pCO_2$: for example, higher solubility given by lower SST, a larger vertical contrast of DIC concentration by the even more stratified ocean (e.g. Kobayashi et al. 2021), and/or larger carbon storage in the deep water by stronger biological pump. The more efficient carbon storage given by these processes would relax the problem of too-high carbonate ion concentrations. Second, the compatibility with the post-glacial shallow water deposition of $CaCO_3$ would need to be satisfied, too. In expLGMss that needed the smallest amount of appended alkalinity of the three LGM experiments, the applied alkalinity corresponded to 2.5e16 mol of $CaCO_3$. This value is already close to the lower limit of the independently-estimated amounts of the shallow water deposition (i.e. 2.2e16 mol) that would have removed alkalinity from the ocean. Therefore, to incorporate the likely post-glacial deposition of $CaCO_3$ and accompanying reduction of alkalinity inventory into the evolution of the climate from the LGM to the modern, another source of alkalinity might be required. More dissolution of $CaCO_3$ in the deep-ocean sediments or more input of alkalinity from the land weathering would be able to serve as the alkalinity source. Considering that the amount of deep-sea carbonate burial is estimated to be rather higher during the last 20krys (Cartapanis et al. 2018), the increased input by the land weathering

might be a more plausible explanation. Future studies to deal with the transient evolution from the LGM are expected to give more insights into this issue. In a revised manuscript, we will extend a discussion section by including these discussion.

Q5: Section 4.2 encompasses many comparisons of the LGM experiments to reconstructions of various parameters. However, I feel that there is a missed opportunity by not visualizing these results to a greater extent (currently only the CaCO3 MAR model-data comparison is shown). One could for instance show the LGM d13C and nutrient data by Oppo et al. (2018) in Figure 3. Further, the [CaCO3] gradients discussed in the text could be plotted against the reconstructions. This would surely help to better demonstrate where the model performs well and where there are biases.

A5: As in A2, we will significantly update the plots, some of which will include visual model-data comparisons as the reviewer suggested. For $\delta^{13}C_{DIC}$, we will overlap dots showing data by Peterson et al. (2014) and Yu et al. (2020). As in the current manuscript, we will refer to Oppo et al. (2018) in the main text because they do not provide point data for Holocene. For the carbonate ion, we will use the data by Yu et al. (2020). As to phosphate, we will not make overlaid plots because Oppo et al. (2018) only provide estimated distributions of phosphate given by inversion and do not have point data. Moreover, for these tracers, we will add plots for a Pacific section.

Q6: The authors try to assess the validity of the DIC – ventilation age relationship used by Sarnthein et al. (2013) and Skinner et al. (2015) and come to the conclusion that it does not hold for the LGM. However, one has to note that the previous studies by Sarnthein et al. (2013) and Skinner et al. (2015) used radiocarbon ventilation ages while in the present study the ideal age of the model was used for the assessment. In this context it is noteworthy that the ideal age and the radiocarbon ventilation age behave quite differently in the (model) ocean, mostly due to the additional effect of limited air-sea gas exchange under sea-ice for radiocarbon that the ideal age tracer does not see. This effect should be much stronger for the LGM simulations than the PI due to the colder temperatures and hence larger sea ice extent. It is therefore possible (or even rather likely) that the radiocarbon ventilation age is much older in the LGM simulations than in PI while the ideal age is younger in the global

mean and the previously proposed relationship still holds. The DIC – age relationship should therefore be reassessed with respect to this issue.

A6: Thank you very much for this insightful comment. We will update the relevant figures and discussion by taking account of the reservoir ages of modern surface water depending on ocean basins based on Matsumoto (2007), and will further modify the LGM counterpart by considering the estimated increase in surface reservoir ages given by Skinner et al. (2017). These modification will alter the intercept of each regression line but will not affect the slope of them. As a result, the fact that the LGM model oceans have a different structure of the DIC–age relationship, where the Atlantic branches are separated from the others, is still valid. Although it is a discussion in a somewhat idealized framework, we consider that the basic ideas behind it (the mixing of northern-sourced water and the DIC-enriched southern-sourced water, and the effect of depth domain for the regression) would be useful for future more detailed examination.

Minor points

Q7: P1, L3-5: This sentence is slightly confusing, considering that you simulated time slices but here argue with the evolution of the reservoirs.

A7: In a revised manuscript, we will modify the initial sentences in the abstract.

Q8: P2, L2: The penultimate interglacial was MIS 7. Do you instead mean the last interglacial (i.e., the Eemian)?

A8: Thank you for pointing out. As the reviewer rightly assumed, we meant the last interglacial. We will modify it in a revised manuscript.

Q9: P2, L5: Orbital configuration not orbital elements.

A9: We will modify the manuscript following this comment.

Q10: P2, L24: As far as I'm aware there are no reconstructed concentrations of DIC.

A10: There we meant the estimate by Sarnthein et al. (2013) that this study had used as a constraint. We will rephrase the part concerned into "the estimated rise of mean DIC concentration in the deep ocean".

Q11: P3, L20: Do you mean that POM was fully remineralized in the bottommost cells?

A11: Yes. We will rephrase "dissolved in the bottom layer" into "remineralized in the bottommost cells".

Q12: P4, L8: Can you give a percentage for the adjusted mean salinity and nutrient concentrations.

A12: We increased the salinity by 1 psu and the other tracers' concentrations by 3%. We will add this information to a revised manuscript.

Q13: P4, L10: Was the 2.5 kyr spinup enough to reach equilibrium?

A13: The $p$CO$_2$ in the atmosphere was superbly equilibrated that the drift in the last 500 model years was 0.6 ppm or less (depending on the simulation). For the other tracers, if we select $\delta^{13}C_{DIC}$ as an example that should give the most strict criteria because of a uniform initial condition, the drift of $\delta^{13}C_{DIC}$ in the deep Northern Pacific (at 30N, 150W, 2900m) was 0.06 permil or less, which was less than a typical magnitude of data uncertainty. Therefore, we judge that the modelled states are reasonably equilibrated.

Q14: P4, L14: The Ruddiman Belt is defined by the deposition of ice rafted debris during Heinrich Events. The reference to this could therefore lead to confusion. Instead, better simply refer to the latitudinal band where the freshwater was applied.

A14: Thank you for pointing out. We will delete the relevant part "so-called 'Ruddiman belt'". The latitudinal band has been already defined in the current version of the manuscript.

Q15: P4, L14: Since the freshwater addition was compensated for, I would try to avoid the word "hosing".

A15: We will replace the word "hosing" with "freshwater forcing".

Q16: P5, L1: Here you mention that the atmospheric d13C signature was prescribed. Isn't this in conflict with freely evolving atmospheric CO2 concentrations in terms of the 13C budget?

A16: Although the treatment of $^{13}C$ in the current model configuration is not theoretically self-consistent as the reviewer pointed out, we took the approach for three practical reasons. 1. The prescribed atmospheric $\delta^{13}C$ that was fixed to a reliable value was expected to contribute to a better model representation of $\delta^{13}C_{DIC}$ in the ocean to be compared with observation-based data. 2. The deviation of the atmospheric $pCO_2$ from the required 190 ppm was minimum, so that the simulated state would reasonably approximate the consistent (with the also-required atmospheric $\delta^{13}C$) LGM state. 3. As far as we recognize, the available carbon isotope package provided by Jahn et al. (2015) does not deal with atmospheric $\delta^{13}C$ that evolves interactively and self-consistently with the air-sea gas exchange in the model, and therefore, further substantial model-development will be needed to implement it, which is beyond the scope of this study.

We will add similar descriptions to these to a revised manuscript.

Q17: P5, L15: It's Atlantic Meridional Overturning Circulation not ocean circulation.

A17: We will correct the error.

Q18: P5, L17-18: The zero isoline of the stream function is not equivalent to the separation of AABW and NADW as can be seen from dye experiments (e.g., for CESM: Gu et al., 2020).

A18: We realized that those two elements are not equivalent to each other, and that is why we used the word "associate". However, we admit the current expression is misleading, and will delete "which can be associated with the North Atlantic Deep Water (NADW) and Antarctic Bottom Water (AABW) masses" in a revised manuscript.

Q19: P5, L21: To me, it appears from Figure 2 that LGMws does not have a stronger bottom circulation than LGM or PI.

A19: We will enhance the current Fig.2 by including common stream-function plots, which will show that expLGMws has a stronger penetration of bottom water up to 20N, which is not resolved by the current plots.

Q20: P5. L26: Directly inferring from roughly correct SST changes the correct atmosphere-ocean partitioning of CO2 is quite a stretch. This completely ignores the other carbon pumps that also play a role in the partitioning.

A20: We only meant the air-sea gas exchange with the "carbon distribution between the atmosphere and the ocean", but we admit that the current expression is misleading. We will delete "hence the carbon distribution between the atmosphere and the ocean" because it is sufficient to mention the reasonable solubility at that point.

Q21: P5, L27: This is in conflict with Figure 3c. However, I suspect that something went wrong in Figure 3c.

A21: We re-plotted the Atlantic sections of salinity and added new horizontal plots as well to confirm the current Fig.3c is correct. Although the current expression "the vertical gradient of salinity is larger in most regions" is still valid theoretically because that is the case in the other oceans than the Atlantic (and in a small part of Atlantic), we will rephrase the sentence because it is misleading and inconsistent with the Atlantic section plot for expLGMws. Instead, we will extend the description of modeled salinity in a revised manuscript by introducing new plots to show comparisons with reconstructed paleo-salinity.

Q22: P5, L31: "In expPI, we obtained 276 ppm". Please rephrase and expand this sentence.

A22: Combined with the next question, we will rephrase the relevant part.

Q23: P6, L1: The model is surely tuned to this PI pCO2, I therefore think that this is not necessarily an indication of the models "excellent ability" to predict pCO2.

A23: Agreeing with the reviewer, we will rephrase the sentences.

Q24: P6, L11: Typo "relfected".

A24: We will correct it.

Q25: P6, L18: Please try to avoid the word "observed" when talking about model results, as it suggests that the finding is derived from observations.

A25: We will modify the relevant parts.

Q26: P6, L25: But the very deep is younger than PI not older and the stream function (Fig. 2) indicates stronger or equal advection in the deep of all LGM runs compared to PI. How does this fit together? Was the longer-lasting storage of organic matter only at the mid-depth between 2 and 3 km? Why is the phosphate concentration also elevated below 3 km? Have you diagnosed the remineralized phosphate fraction from the model?

A26: We appreciate this beneficial remark. We have diagnosed the fraction of remineralized phosphate by estimating the remineralized amount based on AOU and the Redfield ratio defined in the model. The fraction is indeed higher in the all depth ranges deeper than 2000 m in the LGM experiments than in expPI, which gives another support for the fact that the increased phosphate concentration were mainly caused by an increase in remineralized phosphate as mentioned in the current version of manuscript. However, as the reviewer pointed out, we found that the reasons for the increased remineralized phosphate described in the current manuscript is insufficient. In the very deep water below 3km, the effect of the increased biological pump in the mid-to-high latitudes of the Southern hemisphere and the northward transport of remineralized nutrients by the bottom circulation overwhelmed the counter-effect of the younger age of the very deep water. We will add these description to a revised manuscript.

Q27: P9, L13-14: As mentioned before, from the ideal age it is clear that the bottom water was in fact not more stagnant for all LGM simulations.

A27: Similarly to A26, we will add words to describe the effect of increased biological pump.

Q28: Figure 3c: The distribution looks rather strange compared to the other experiments and other tracers. Please double-check.

A28: Please see A21. We will add discussion about the influence of the freshwater forcing on the simulated salinity.

Q29: Figure 6: Most of the map is white. Does that mean that in ~80% of the grid cells the 1°x1° bathymetry is outside the POP2 depth domain and no CaCO3 MAR can be calculated? If yes, can this be improved to show a continuous map?

A29: Motivated also by a comment from the other reviewer, for a revised manuscript we coupled MEDUSA simply or directly to every bottom grid cell of POP2 as in the preceding study (Kurahashi-Nakamura et al., 2020) to involve the whole ocean floors and carried out similar experiments again. This new method has enabled us to provide global continuous maps of MAR covering the entire ocean. Moreover, the global burial amount of $CaCO_3$ will be available with the new method.

References:

Cartapanis, O., Galbraith, E. D., Bianchi, D., and Jaccard, S. L.: Carbon burial in deep-sea sediment and implications for oceanic inventories of carbon and alkalinity over the last glacial cycle, Climate of the Past, 14, 1819–1850, https://doi.org/10.5194/cp-14-1819-2018, 2018.

Jahn, A., Lindsay, K., Giraud, X., Gruber, N., Otto-Bliesner, B. L., Liu, Z., and Brady, E. C.: Carbon isotopes in the ocean model of the Community Earth System Model (CESM1), Geoscientific Model Development, 8, 2419–2434, https://doi.org/10.5194/gmd-8-2419-2015, 2015.

Kobayashi, H., Oka, A., Yamamoto, A., and Abe-Ouchi, A.: Glacial carbon cycle changes by Southern Ocean processes with sedimentary amplification, Science Advances, 7, eabg7723, https://doi.org/10.1126/sciadv.abg7723, 2021.

Matsumoto, K.: Radiocarbon-based circulation age of the world oceans, J. Geophys. Res., 112, C09004, doi:10.1029/2007JC004095, 2007.

Oppo, D. W., Gebbie, G., Huang, K.-F., Curry, W. B., Marchitto, T. M., and Pietro, K. R.: Data Constraints on Glacial Atlantic Water Mass Geometry and Properties, Paleoceanography and Paleoclimatology, 33, 1013–1034, https://doi.org/10.1029/2018PA003408, https://agupubs.onlinelibrary.wiley.com/doi/abs/10.1029/2018PA003408, 2018.

Peterson, C. D., Lisiecki, L. E. and Stern, J. V.: Deglacial whole-ocean d$^{13}$C change estimated from 480 benthic foraminiferal records, Paleoceanography, 29, 549–563, doi:10.1002/2013PA002552, 2014.

Sarnthein, M., Schneider, B., and Grootes, P. M.: Peak glacial $^{14}$C ventilation ages suggest major draw-down of carbon into the abyssal ocean, Climate of the Past, 9, 2595–2614, https://doi.org/10.5194/cp-9-2595-2013, 2013.

Skinner, L., McCave, I., Carter, L., Fallon, S., Scrivner, A., and Primeau, F.: Reduced ventilation and enhanced magni- tude of the deep Pacific carbon pool during the last glacial period, Earth and Planetary Science Letters, 411, 45–52, https://doi.org/https://doi.org/10.1016/j.epsl.2014.11.024, https://www.sciencedirect.com/science/article/pii/S0012821X1400716X, 2015.

Skinner, L. C., Primeau, F., Freeman, E., de La Fuente, M., Goodwin, P. A., Gottschalk, J., Huang, E., McCave, I. N., Noble, T. L., and Scrivner, A. E.: Radiocarbon constraints on the glacial ocean circulation and its impact on atmospheric $CO_2$, Nature Communications, 8, 16010, https://doi.org/10.1038/ncomms16010, 2017.

Yu, J., Menviel, L., Jin, Z. D., Anderson, R. F., Jian, Z., Piotrowski, A. M., Ma, X., Rohling, E. J., Zhang, F., Marino, G., and Mc- Manus, J. F.: Last glacial atmospheric $CO_2$ decline due to widespread Pacific deep-water expansion, Nature Geoscience, 13, 628–633, https://doi.org/10.1038/s41561-020-0610-5, 2020.

---

## Referee Report (RR1)

**Review of cp-2022-8**

**Glacial state of the global carbon cycle: time-slice simulations for the last glacial maximum with an Earth-system model**

**by T. Kurahashi-Nakamura, A. Paul, U. Merkel, and M. Schulz**

The authors nicely improved the manuscript in this revised version. I confirmed that the points raised in the previous review round have been improved and well discussed. In particular, the comparison between models and data using various tracers for the LGM time slice is valuable. Some minor points are noted below.

**General comments:**

This study conducted a two-step spin-up to establish values of DIC and alkalinity that are compatible with glacial ocean restoration. Additional charts or more detailed explanations would make it easier to understand the intent of the method.

Would you evaluate how much atmospheric pCO2 would be obtained in each LGM experiment if there were no additional increments in DIC and alkalinity, which may come from changes in shallow water deposition of CaCO3? It may support the significance of continental shelf processes.

**Specific comments:**

P5/L27: It may be helpful to clarify the additional increase of 100 μmol kg-1 in this sentence.

P8/L16: Is AOU calculated explicitly in the model? If not, would you indicate how it is calculated?

P8/L21: What caused the positive anomaly of d13C in the North Pacific in expLGMws. From Fig 7e, it is assumed that this is due to stronger volume transport from the Southern Ocean, which results in a smaller effect of remineralization.

P10/L15: Is the small MAR of CaCO3 in all experiments in the Southern Ocean due to the dominance of other particle fluxes such as opal?

P10/L21 In Discussion section 4.1, the authors provided changes in the budget of oceanic DIC and alkalinity between the LGM and modern. It would be easier to understand if the estimated fluxes shown here could be visualized.

Figures: There are abbreviations in the title of figures that are not explicitly stated (e.g. IFRAC). Also, it would be better to correct 330E and 210E in the title of the figure to 30W and 150W, respectively.

---

## Author Response (AR2)

[Reply to Reviewer #1]

We appreciate the positive evaluation of the first revision and the additional comments by the reviewer.

Q1: This study conducted a two-step spin-up to establish values of DIC and alkalinity that are compatible with glacial ocean restoration. Additional charts or more detailed explanations would make it easier to understand the intent of the method.

A1: A chart (Fig. 2) was added to illustrate the workflow.

Q2: Would you evaluate how much atmospheric pCO2 would be obtained in each LGM experiment if there were no additional increments in DIC and alkalinity, which may come from changes in shallow water deposition of CaCO3? It may support the significance of continental shelf processes.

A2: We had also carried out three more LGM simulations without the additional alkalinity, which had resulted in 20 to 43 ppm higher $p\mathrm{CO_2}$ depending on simulations. These simulations depict ocean states that satisfy the constraint of deep-ocean carbon reservoir but lacking the contribution of alkalinity increase. If DIC increase corresponding to the $\mathrm{CaCO_3}$ deposition was removed as well, the simulated $p\mathrm{CO_2}$ would be somewhat lower than the values shown above. These facts suggest that the shallow-water deposition of $\mathrm{CaCO_3}$ would only explain a minor portion of the $p\mathrm{CO_2}$ increase on the deglaciation as suggested by Ridgwell et al., (2003). We added a similar description to the manuscript (section 4.2).

Q3: P5/L27: It may be helpful to clarify the additional increase of 100 μmol kg-1 in this sentence.

A3: We modified the sentence to include the information.

Q4: P8/L16: Is AOU calculated explicitly in the model? If not, would you indicate how it is calculated?

A4: Yes, AOU is explicitly calculated in the model. We added a description to Section 2.1.

Q5: P8/L21: What caused the positive anomaly of d13C in the North Pacific in expLGMws. From Fig 7e, it is assumed that this is due to stronger volume transport from the Southern Ocean, which results in a smaller effect of remineralization.

A5: Thank you for the useful discussion, and we agree with the reviewer's opinion. A relevant description was added to the text.

Q6: P10/L15: Is the small MAR of CaCO3 in all experiments in the Southern Ocean due to the dominance of other particle fluxes such as opal?

A6: Yes. The small MAR of $CaCO_3$ in the Southern Ocean results from low $CaCO_3$ productivity compared to opal fixation, as the reviewer assumed. We added a corresponding sentence to section 3.6.

Q7: P10/L21 In Discussion section 4.1, the authors provided changes in the budget of oceanic DIC and alkalinity between the LGM and modern. It would be easier to understand if the estimated fluxes shown here could be visualized.

A7: We added a figure (Fig. 12) to summarize the discussion in Section 4.1.

Q8: Figures: There are abbreviations in the title of figures that are not explicitly stated (e.g. IFRAC). Also, it would be better to correct 330E and 210E in the title of the figure to 30W and 150W, respectively.

A8: We modified the figures accordingly.

[Reply to Reviewer #2]

We appreciate the positive evaluation of the revised manuscript and the additional comments by the reviewer.

Q1:

P1, L20: 'was characterized' not 'is characterized'.

P3, L5: 'inflow' and 'outflow' sound awkward in this context and following. Instead, I recommend 'input' and 'removal'.

P4, L2: Replace 'increment' with 'increase' (also in the abstract).

P4, L16: 'two combined factors' instead of 'combined two factors'.

P5, L33: Please add a reference for the standard CESM1.2 parameters.

P6, L14: Please add a reference to Fig. 1 here.

P12, L31: 'on average' not 'in average'.

P14, L30: 'added' instead of 'appended'.

P15, L29: Change 'influences on the ...' to 'influences the …'.

A1: We modified the manuscript according to these comments.

Q2: Fig. 1: Add 'constant' to the last sentence of the caption (i.e., "to keep the total volume of sea water constant.")

A2: Corrected.

Q3: Fig. 2: To better distinguish both overturning cells in panels a-d it would be beneficial to either mark the zero-isoline with a thicker line or change the contour lines of the negative values to dashed lines.

A3: We changed the filling color of the contours to better show the structures.